# Frontal destabilisation of Stonebreen, Edgeøya, Svalbard

Tazio Strozzi[1], Andreas Kääb[2], Thomas Schellenberger[2]

[1]Gamma Remote Sensing, Worbstrasse 225, 3073 Gümligen, Switzerland

[2]Department of Geosciences, University of Oslo, P.O. Box 1047, Blindern, 0316 Oslo, Norway

*Correspondence to: Tazio Strozzi (strozzi@gamma-rs.ch)*

**Abstract.** In consideration of the strong atmospheric warming that has been observed since the 1990s in polar regions there is a need to quantify mass loss of Arctic ice caps and glaciers and their contribution to sea level rise. In polar regions a large part of glacier ablation is through calving of tidewater glaciers driven by ice velocities and their variations. The Svalbard region is characterized by glaciers with rapid dynamic fluctuations of different types, including irreversible adjustments of

calving fronts to a changing mass balance and reversible, surge-type activities. For large areas, however, we do not have much past and current information on glacier dynamic fluctuations. Recently, through frequent monitoring based on repeat optical and SAR satellite data, a number of zones of velocity increases have been observed at formerly slow-flowing calving fronts on Svalbard. Here we present the dynamic evolution of the southern lobe of Stonebreen on Edgeøya. We observe a slowly steady retreat of the glacier front from 1971 until 2011, followed by a strong increase in ice surface velocity along

with a decrease of volume and frontal extension since 2012. The considerable losses in ice thickness could have made the tide-water calving glacier, which is grounded below sea level some 6 km inland from the 2014 front, more sensitive to surface melt-water reaching its bed and/or warm ocean water increasing frontal ablation with subsequent strong multi-annual ice-flow acceleration.

## 1 Introduction

Mass loss from glaciers and ice sheets contributes about one third to current sea level rise (Church et al., 2013). In polar regions, a large part of glacier mass loss is through frontal ablation of tidewater glaciers (Rignot et al., 2008; Blaszczyk et al., 2009; Burgess et al., 2013; Khan et al., 2015; McNabb et al., 2015), which includes calving of icebergs and frontal melt. Related calving fluxes and their changes over time are mainly driven by lateral ice velocities and their variations. The increase of archived and newly acquired optical and Synthetic Aperture Radar (SAR) satellite data available for quantifying

ice surface flow together with progress in methods in particular for handling large satellite data sets with increasing temporal resolution open up for new possibilities to monitor ice flow over large regions, and to detect and understand related changes. Ice mass fluctuation in Svalbard is similarly controlled by both dynamic and surface mass balance change, making glacier

dynamics an important factor of glacier's mass turnover and change (Blaszczyk et al., 2009; Moholdt et al., 2010; Dunse et al., 2015). The total calving flux of Svalbard is dominated by a few large and fast-flowing glaciers ( Dowdeswell et al., 2008), and variations in their speed can thus have large influence on the total mass balance of the archipelago's glaciers. So far, two main types of fast-flowing glaciers have been described on Svalbard. A few glaciers, such as Kronebreen and

Kongsbreen near Ny Ålesund, Western Spitsbergen, are continuously fast-flowing, with maximum speed of more than 2-3 m/day at the calving front. These glaciers show seasonal variations in speed related to meltwater input to the glacier base (Dunse et al., 2012; Schellenberger et al., 2015). Also, increases in overall speed are observed for these glaciers together with the retreat of calving fronts due to a combination of general glacier mass loss, overdeepenings of the glacier bed, warm ocean water reaching the fronts and causing retreat and reduced butressing (Luckman et al., 2015), and changes in the back-

stressing sea ice cover in front of the glaciers. On the time-scales of, at least, decades such calving front retreats are typically irreversible until a considerable mass gain enables the glacier to re-advance from one pinning-point to the next.

A second, and very important type of fast-flowing glaciers on Svalbard are surging glaciers (Murray et al. 2003). Surge-type glaciers undergo a cyclic behaviour, with periods of rapid acceleration and advance (active phase) followed by periods of

slow flow where ice fluxes are less than balance fluxes (quiescent phase) (Clarke, 1987). In a typical Svalbard glacier surge cycle the surge starts with a years-long period of steady acceleration, followed by a months-long period of relatively rapid acceleration, a length of the active phase of typically 3-10 years, and a very gradual end of the fast flow phase with velocity decreasing over a years-long period (Murray et al., 2003). The long active (~7-15 years) and quiescent (~50-100 years) phases, combined with surge termination that occurs over a multi-year period and velocity changes between the two phase of

one or two orders of magnitude, suggest that the Svalbard type surges are linked to changes in basal thermal conditions rather than subglacial water pressure (Murray et al., 2003). Surging glaciers are able to temporarily discharge large ice masses into the ocean and thus vary the total glacier mass balance of the ice masses of the Svalbard Archipelago considerably over shorter time scales. Two recent prominent surges are the ones of Nathorstbreen in Southern Spitsbergen and Basin-3 on Austfonna. The Nathorstbreen glacier system started to surge in 2009 and an advance of about 15 km was

observed (Sund et al. 2014). The northern branch of Basin-3 showed a stepwise acceleration over multiple years at least

since 2008, which ended in a basin-wide surge starting in 2012 with velocities up to 20 m/day (McMillan et al., 2014, Dunse et al., 2015). The impact of climatic changes on surges on Svalbard is debated, but Dunse et al. (2015) suggest a hydro-thermodynamic feedback mechanism where increased meltwater production and input to the glacier bed is able to stepwise trigger a surge-type instability. Glacier mass losses due to surges are in principle reversible once the glacier stops its fast discharge and is thus able to replenish its mass through accumulation. Also, transitions between surge-type behaviour and more continuous fast-flow seems possible such as for Monacobreen (Schellenberger et al, 2016).

Recently, through frequent monitoring based on repeat optical and SAR satellite data, a number of zones of speed increases have been observed at formerly slow-flowing calving fronts on Svalbard, and questions arise whether these are forms of irreversible adjustments of calving fronts to a changing mass balance, or parts of reversible surge-type activities. For instance, a frontal zone of Basin-3 of Austfonna that accelerated spatially separated from the main glacier stream was later incorporated in the surge once it reached its full extent, and thus it was viewed as part of the Basin-3 surge (Dunse et al., 2015). Here, we investigate a zone of recent acceleration of a formerly very slow flowing calving front of Stonebreen on Edgeøya, Eastern Svalbard (23.8 E, 77.8 N), in order to characterise its spatio-temporal dynamic pattern and the potential nature and significance of the instability. As a secondary goal, we also evaluate the potential of frequent standard acquisitions by new Earth observation missions such as Sentinel-1 (radar) and Landsat 8 (optical) to detect and analyse such spatially limited and temporally variable glacier instabilities.

## 2 Study site

Edgeøya is located in the southeast of the Svalbard archipelago and is 5,073 km$^2$ in area, making it the third largest island in the Svalbard archipelago. The eastern side of Edgeøya is covered by the Edgeøyjøkulen ice cap, which had an extension of 1365 km$^2$ in 1985 (Dowdeswell and Bamber, 1995). The ice cap on Edgeøya, together with its neighbour ice cap on Barentsøya, is among the least well known in Svalbard (Dowdeswell and Bamber, 1995). To our best knowledge, there are no existing field studies of either the ice thickness or the mass balance of these glaciers and ice caps. The tidewater ice cliffs of eastern Edgeøya are over 80 km long and produce small tabular icebergs (Dowdeswell and Bamber, 1995). With the

exception of the surge-related advances, the tidewater ice masses of eastern Edgeoya have been in retreat over the Twentieth Century (Nuth et al., 2013, Arendt et al., 2015).

Several of the ice-cap outlet glaciers on Edgeøya are interpreted to be of surge-type, based on a combination of direct observations and analysis of vertical and oblique aerial photographs (Liestøl, 1993). Stonebreen (Fig. 1) is the largest glacier on the ice cap with an area of 687 km$^2$ in 1971 (Nuth et al., 2013) and of 582 km$^2$ in 2006-2007 (Arendt et al., 2015). The northern lobe of Stonebreen appears to have surged between 1936 and 1971. However, the southern lobe investigated here does not appear to have surged during this period, indicating that different basins within this ice cap behave as dynamically separate units (Dowdeswell and Bamber, 1995).

Airborne radio-echo sounding at 60 MHz over the ice masses of Edgeøya has provided ice thickness and elevation data (Dowdeswell and Bamber, 1995). Ice is grounded below sea level up to about 20 km inland from the tidewater terminus of the northern lobe of Stonebreen. For the southern lobe of Stonebreen investigated here, ice seemed to be grounded below sea level some 8 km inland in 1986 (Dowdeswell and Bamber, 1995), corresponding to 6 km from the 2014 front. Ice thickness is from <100 m close to the margins to about 250 m in the interior of Edgeøyjøkulen. The ice masses on Edgeøya are believed to be at the pressure melting point with a basal hydrological system (Dowdeswell and Bamber, 1995).

## 3 Data and Methods

### 3.1 Coastal Outlines

Coastal outlines were manually digitized from Landsat imagery from 1994 to 2015 with additional information from the Randolph Glacier Inventory (RGI) 5.0 (Arendt et al., 2015) and vertical aerial photograph in 1971 (Nuth et al., 2013). Acquisition dates and sensors name of the satellite optical imagery used for the mapping of the coastal outlines are indicated in Table 1. The primary data set used for RGI 5.0 is SPOT5 orthoimages at 5m resolution from 2007-2008 (Arendt et al., 2015; Nuth et al., 2013). Landsat data are available orthorectified from the U.S. Geological Survey and were not co-registered to a common geometry but relative co-registration between the scenes used was checked as well as absolute georeferenced against mapped rock outcrops and non-glaciated coastlines. Only Landsat scenes that passed these visual

checks were used further. All images are from the summer months with good contrast on clean ice. The relatively uncertainty of glacier delineations is on the order of one pixel, i.e. up to 30 m.

## 3.2 Glacier Elevation Change

Three Digital Elevation Models (DEM) were available for our analysis: the Norwegian Polar Institute DEM (NPI DEM), the TanDEM-X Intermediate DEM (IDEM), and an ASTER DEM (AST14DEM).

The NPI DEM (Norwegian Polar Institute, 2014) is based on 1:100 000 scale topographic maps derived from aerial photography in 1970. It is provided at 20 m posting in UTM projection for zone 33N and the WGS 1984 spatial reference system. From a study on the nearby Digerfonna ice cap (just west of Edgeøyajokulen) we estimate the accuracy of the elevations of the NPI DEM to be on the order of few meters to around ±12m for difficult terrain (Kääb 2008). The IDEM (DLR EOC, 2013) is based over Edgeøya on TanDEM-X acquisitions from 12/12/2010 to 26/03/2012. It is provided in 3 arcsec geographic coordinates, with a posting corresponding to approximately 90 m. The indicated absolute horizontal and vertical accuracies are < 10 m (DLR EOC, 2013). Independent tests performed with a TanDEM-X DEM of Mount Etna (Italy) indicated that the difference of the elevations provided by TanDEM-X with those measured with GPS over more than 100 benchmarks are 0.7 m with a standard deviation of 5.2 m (Wegmüller et al., 2014). For an ASTER satellite stereo scene of 15/07/2014 we acquired an AST14DEM product (LPDAAC, 2016). The product is provided with a posting of 30 m. Over the nearby Digerfonna Ice Cap Kääb (2008) found an accuracy of the AST14DEM of ±12 m RMS or better.

Glacier elevation change is computed by subtraction of the IDEM from the NPI DEM and of the AST14DEM from the IDEM after resampling of all DEMs to 100 m posting on the UTM 33 projection. Visual inspection suggested that lateral co-registration between the three DEMs was not necessary and would have little to no effect, among others because the slope of the lower part of the Stonebreen studied here is only about 1°-2° steep (Nuth and Kääb, 2011). Vertically, all DEM differences were checked over stable terrain and an offset of 40 m was corrected for the AST14DEM.

## 3.3 Ice Surface Velocity

We analysed a series of satellite SAR images acquired by the ERS-1, ERS-2, ALOS PALSAR, Radarsat-2, and Sentinel-1 missions from 1994 to 2016 and of optical Landsat 8 images from 2014 to 2016. Acquisition dates and time intervals of the satellite image pairs considered in our study are given in Supplementary Table S1.

5 An ERS-1 image pair of 1994 and an ERS-2 image pair of 2011 from the 3-day repeat campaigns were processed to differential SAR interferograms with use of the IDEM (Bamler and Hartl, 1998; Rosen et al., 2001). The acquisition date of the IDEM matches that of the ERS-2 data and we do not expect any major topographic signal left on the differential interferogram. On the other hand, important ice surface elevation changes (> 100 m) occurred between the date of the IDEM (2010-2013) and that of the ERS-1 data (1994), but because the perpendicular baseline of the ERS-1 pair is only 15 m, the 10 phase artefacts are small (i.e. $0.32\pi$ or 57º for a height error of 100 m) (Strozzi et al., 2001). Therefore, on both ERS-1 and ERS-2 differential SAR interferograms the phase signals can be interpreted as ice surface displacement in the satellite line-of-sight direction with possible atmospheric disturbances. Phase unwrapping to obtain displacement values (Werner et al., 2002) was not attempted because undersampling of the SAR data in relationship to the rate of movement is easily causing phase unwrapping errors, similar to what is observed in the case of mining (Spreckels et al., 2001, Przyłucka and al., 2015) 15 or rockglaciers (Barboux et al., 2015). Processing of the ERS-1 and ERS-2 3-day data with offset-tracking procedures (Strozzi et al, 2002; Paul et al., 2015) did not produce useful results, because a 1/20th of pixel precision in offsets estimation over a 3 days repeat period yields a displacement error of about 100 m/yr, which is larger than the velocity observed over Stonebreen in past years.

ALOS PALSAR Fine-Beam Single (FBS) image pairs in 2010 and 2011, Radarsat-2 Wide  image pairs from 2009 to 2016, 20 Radarsat-2 Wide Ultra-Fine (WUF) image pairs in 2016, and Sentinel-1 Interferometric Wide Swath (IWS) SAR  image pairs in 2015 and 2016 were processed with offset-tracking procedures (Strozzi et al, 2002; Paul et al., 2015) to three-dimensional ice surface displacement maps combining the slant-range and azimuth offsets by assuming that flow occurs parallel to the ice surface estimated from the DEM (e.g. Mohr et al., 1998). Matching window sizes of 64x196, 30x120, 128x128 and 512x128 pixels were applied to the ALOS PALSAR, Radarsat-2 Wide, Radarsat-2 WUF and Sentinel-1 IWS 25 data, respectively. Mis-matches or blunders were filtered by applying a threshold on the correlation coefficient, by iteratively

discarding matches based on the angle and size of displacement vectors in the surrounding area, and by using a high-pass filter on the resulting displacement fields (Paul et al., 2015). The error in the estimation of ice surface velocity with ALOS PALSAR data separated by a repeat-cycle of 46 days is on the order of 10 m/yr (Paul et al., 2015). For a repeat-cycle of 92 days a similar error is expected, although the spatial coverage with valid information is reduced. The expected displacement

error of the Radarsat-2 Wide, Radarsat-2 WUF and Sentinel-1 IWS data was estimated by assuming a precision of 1/20th of a pixel in the offset estimation. We estimate for (i) Radarsat-2 Wide data with pixel sizes in ground-range and azimuth direction of about 20m x 5m and a time interval of 24 days a displacement error of ~15 m/yr, for (ii) Radarsat-2 WUF data with pixel sizes in ground-range and azimuth direction of about 3m x 3m and a time interval of 24 days a displacement error of ~5 m/yr, and (iii) for Sentinel-1 IWS data with pixel sizes in ground-range and azimuth direction of 8m x 20m and a time

interval of 12 days a displacement error of ~30 m/yr.

Ice velocities from repeat Landsat 8 data acquired in the summer of 2014, 2015 and 2016 were measured to fill temporal gaps in the SAR-derived velocity time series using standard normalized cross-correlation methods (Kääb and Vollmer, 2000; Debella-Gilo and Kääb, 2011; Heid and Kääb, 2012). In this study, matching window sizes of 15 pixels were applied based on Landsat 8 pan band data (15 m resolution). For areas of good visual contrast, such as those in our study site due to

crevassed and snow-free glacier surfaces, displacement accuracies of 10-20% of a pixel (15 m) can be reached (Heid and Kääb, 2012), i.e. 1.5-3 m corresponding to 24-48 m/yr for a time interval of 16 days. For time intervals different than the optimal factor of 16 days nominal repeat cycle or integer multiples of it, displacement accuracies are potentially reduced due to differential orthorectification errors (Kääb et al., 2016). We minimized these effects by choosing scenes where Stonebreen is close to the satellite ground track, and carefully checked the scenes by flickering the repeat images and the resulting

velocity vectors for offsets in cross-track direction. Further, flow direction is not perpendicular to the Landsat track direction which further reduces the effect of orthorectification offsets. In sum, differential orthorectication offsets should affect our Landsat-derived velocities only to a minor extent and not the conclusions drawn from them. Co-registration of the matching scenes was also checked for stable ground and did not reveal any statistically significant offsets. Mis-matches were removed using the same approach as for SAR results (Paul et al., 2015).

# 4 Results

## 4.1 Coastal Outlines

The coastal outlines from the vertical aerial photograph in 1971, the Landsat imagery in 1976, 1994, 2011 and 2015, and RGI 5.0 of the years 2006 and 2007 are shown in Fig. 2. From the aerial and satellite imagery we observe a prominent retreat of all glaciers along the eastern coast of Edgeøya. The maximal retreat of the front of Stonebreen from 1971 to 2011 was larger than 3 km. However, from 2011 to 2015 we observe an advance of almost 500 m of the front of Stonebreen at the centre of Fig. 2.

## 4.2 Glacier Elevation Change

Glacier elevation changes were computed between the NPI DEM of 1970 and the IDEM of 2010/2012 and the IDEM and the ASTER DEM of 2014 and are presented in Figs. 3a and 3b, respectively. From 1990 to 2010/2012 we observe a general pattern of height losses along the coast of Stonebreen. Over the southern lobe the ice surface losses were up to 150 m along the coast. Between 2010/2012 and 2014 we observe over an area of about 15 km$^2$ in the inland of Stonebreen ice surface losses of 50 to 70 m.

## 4.3 Ice Surface Velocity

ERS differential SAR interferograms and ALOS PALSAR, Radarsat-2, Landsat 8 and Sentinel-1 ice surface velocity maps are presented in Figs. 4 to 8. Slow-flowing ice becoming dynamically active is identified over the southern lobe of Stonebreen. In the ERS-1 differential SAR interferogram of 1994 (Fig. 4a) two small dynamically active sectors of about 4 km$^2$ each are identified. The line-of-sight velocities are on the order of 12 (~4 fringes of 2.8 cm each) to 20 cm (~6 fringes) in 3 days, i.e. 15 to 25 m/yr, or 40 to 60 m/yr on a horizontal plane taking into account the 23° incidence angle of the ERS-1 SAR sensor. In the ERS-2 differential SAR interferogram of 2011 (Fig. 4b) a single dynamically active sector of about 14 km$^2$ is visible. The velocity is lower towards the south, where it is approaching 14 cm (~5 fringes) in the line-of-sight direction or 45 m/yr on a horizontal plane. The northern sector is decorrelated, i.e. flowing faster.

ALOS PALSAR data of the same period as ERS-2 SAR (Fig. 5a) quantifies the ice surface velocity in the northern to maximum 300 m/yr. The size of the dynamically active sector determined with ALOS PALSAR is similar to that observed with ERS-2 SAR. Radarsat-2 results of late 2011, i.e. the following winter season (Fig. 5b), indicate a larger dynamically

active sector with  higher velocities up to more than 500 m/yr. Ice velocity maps at higher spatial resolution from Landsat 8 are presented in Fig. 6a for the the summer of 2014 and in Fig. 6b for the summer of  2015. The size of the dynamically active sector increased steadily due to an inland migration of a front of elevated velocities. Maximal velocities are over 1,500 m/yr in both years. The coverage with valid ice surface velocity data are restricted to crevassed fast-flowing glacier sections.

In order to retrieve recent winter ice velocity data we used data of the Sentinel-1 SAR sensor. In the winter of 2015 Sentinel-1 data (Fig. 7a) quantified the size of the dynamically active sector to about 50 km$^2$ with maximal velocities approaching 600 m/yr. After three 12-day campaigns performed in January and February 2015 with nearly identical results over Stonebreen, no further Sentinel-1 data are available until mid of August 2015, when Sentinel-1 acquisitions over Svalbard started on a regular 12-days basis. The Sentinel-1 tracking results with winter images have a better spatial coverage with valid

information, in particular over the interior of the ice cap, but the coverage with valid information over the dynamically active sector of Stonebreen is very good also for the summer data (Fig. 7b). Results are similar to those obtained with Landsat 8 data of the summer of 2015 (Fig. 6b), although at lower spatial resolution and with a less complete picture of glacier speed.

In the newest Sentinel-1 results of the winter of 2016 (Fig. 8a) we observed that the size of the dynamically active sector is increasing again inland due to a further migration of the front of elevated velocities. Almost coincident Radarsat-2 WUF data

(Fig. 8b) with a ground resolution of about 3 m can be used for validation of the Sentinel-1 IWS data. The standard deviation of the difference of ice surface velocity over Stonebreen between Radarsat-2 WUF and Sentinel-1 IWS data are about 50 m/yr. On ice free regions the standard deviations of the Radarsat-2 WUF and Sentinel-1 IWS displacements are between 5 to 10 m/yr and 20 to 30 m/yr, respectively.

A time series of ice surface velocities close to the front of the southern lobe of Stonebreen (716080 E / 8646230 N) is

presented in Fig. 9. Thanks to the frequent monitoring based on repeat optical and SAR satellite data it is possible to well follow the speed-up of Stonebreen from 2009 to 2015, with large seasonal fluctuations. The maximal velocity was reached in October 2015 with values approaching 2,500 m/yr, followed by a decrease down to 500 m/yr in April 2016. The different SAR and optical satellite sensors complement each other very well.

## 5 Discussion

Over the southern lobe of Stonebreen we observe a slowly steady retreat of the glacier from 1971 to 2011 followed since 2012 by a strong increase in ice surface velocity with prominent seasonal variations along with a decrease of volume and an advance in frontal extension. The acceleration phase of the southern lobe of Stonebreen was lasting at least 3 years, more than the months-long period of relatively rapid velocity increase of a typical Svalbard glacier surge cycle (Murray et al., 2003), and the acceleration was not constant but seasonally modulated. So far, no distinct deceleration phase is observed over the glacier. The following processes or their combination could be involved in the instability observed since 2012 and are thus discussed in more detail:

→ increase in surface slope and related increment in driving stress;

→ decrease in ice thickness and related reduction of basal drag;

→ increased surface melt-water input to the glacier bed and related increase in basal pressure;

→ increased frontal ablation due to warm ocean water.

Our interpretation of the observed instability is complicated by the lack of detailed bathymetry in the front of Stonebreen, because water depth is an important factor for the stability and behaviour of tidewater calving glaciers (e.g., Van der Veen, 1996; Vieli et al, 2001). To our best knowledge, only the Norwegian Sea Navigation Chart (2016) gives approximate indications of water depths in front of the southern lobe of Stonebreen. Approximate values are around 25 m (Fig. 1).

### 5.1 Increase in slope

Over the 6-7 km of the current instability (measured in ice flow direction), which roughly coincides with the glacier section that could be grounded below sea level, the surface slope amounted to only ~ 1° in the 1970s NPI DEM. The elevation loss at the glacier front from the 1970s (NPI DEM) to 2010-2012 (IDEM) led to an increase in surface slope of ~ 2°. Assuming a glacier depth of 200 m (Dowdeswell and Bamber, 1995) and an infinite slab with standard parametrization (Cuffey and Paterson., 2010), this increase in slope would have increased ice deformation speeds at the surface from a few dm/yr to a few m/yr. Both these speeds are lower than the increase in surface speeds observed, pointing to a significant contribution of basal sliding even before the instability. At the upper end of the current instability, where large elevation losses of up to 70 m are found between the 2010-2012 IDEM and the 2014 ASTER DEM (Fig. 3b), the surface slope increased from ~ 1.5° to ~ 3.5°,

which could have caused an increase in deformation velocity from ~ 1 m/yr to almost ~ 20 m/yr at the surface. Although hardly being a main reason behind the increase in flow speed, the increase in slope could be involved in a basal feedback mechanism.

## 5.2 Reduction in ice thickness

In 1970 (NPI DEM) the southern lobe of Stonebreen was around 80 m high above sea level. During 2010-2012 (IDEM) it was only 35 m high above sea level. At the location of the 2010-2012 front surface elevation in 1970 was around 110 m. This is a considerable reduction of ice thickness. However, the current glacier thickness should still be far over the floatation level that would only be reached for several hundreds of metres water depth, which is very unlikely. Though, the loss in ice thickness - from ~ 135 m in 1970 to ~ 60 m in 2010/2012 if we assume 25 m water depth at the 2010/2012 location of the front of the southern lobe of Stonebreen - will have caused a significant decrease in basal pressure and thus basal drag. In Fig. 9 we observe that the instability started around 2011, when most of the ice thickness loss should already have happened. This suggests that reduction of ice thickness is not a result of the increase in flow and discharge to the ocean, but rather a cause or part of a combination of causes. However, even if a connection between the thickness losses and the increase in ice flow is not unlikely, we can in theory not rule out from the data we have collected that the thickness changes are independent from the change in flow.

## 5.3 Increased surface melt-water input

The strong seasonal component of the increase in surface speed (Fig. 9), with repeated maxima around September of each year since 2012, suggests a link between surface-melt and multi-annual ice-flow acceleration instead or in combination with the above potential effects from increase in slope or decrease in ice thickness. The variation in speed in Fig. 9 has considerable similarity with the step-wise acceleration of Basin-3 on Austfonna (Dunse et al. 2012; Dunse et al., 2015), where surface speeds fell back only to a higher speed level after summer speed maxima. Dunse et al. (2015) identified an annual hydro-thermodynamic feedback that successively mobilizes stagnant ice regions, initially frozen to their bed, thereby facilitating fast basal motion over an expanding area. Also for Stonebreen the annual increase in speed starts in July with strong increase in production of surface melt water in the region, which is well visible from a strong reduction of the

backscattering intensity on Sentinel-1 Extended Wide Swath (EWS) images (Fig. 10). The time-series of the SAR backscattering coefficient in Fig. 10 was computed using Sentinel-1 EWS images from various orbital configurations to increase the temporal sampling. Ground Range Detected (GRD) data were considered applying a radiometric calibration but not a correction for the different incidence angles. The capability of mapping wet snow and ice conditions by means of SAR data is well proven (e.g. Strozzi et al., 1999) and is due to vanishing background contribution because of increasing absorption and reflection of the incident radiation by liquid water. The speed maxima seem to be reached a bit later (around one month) on Stonebreen than on Basin-3, though the remote sensing measurements on Stonebreen are less resolved than the automatic GPS and 11-day repeat data from TerraSAR-X available for Basin-3. In addition, in contrast to Basin-3 winter velocities over Stonebreen are low, in particular in 2016.

## 5.4 Increased frontal ablation

Besides the impact of surface melt-water reaching the glacier bed, a second process inducing seasonal variations could be seasonal changes in frontal ablation. For Kronebreen in western Spitsbergen the influx of warm ocean water was recently suggested to have triggered a decoupling of the calving front from its former  pinning-point, associated increases in surface speed, and retreat until a new  pinning-point is reached (Luckman et al. 2015; Schellenberger et al., 2015). No data are available to us about changes in ocean temperatures in front of Stonebreen or the recent influx of warm water. However, the fact that the front of the section of Stonebreen investigated here is currently advancing by several hundreds of meters every year rather than retreating does not support the hypothesis that the increase in speed is due to adjustment of the glacier to a new  pinning-point further inland. Though, the influx of warm ocean water could lead to seasonally increased frontal ablation and subsequent reduction of frontal backstress and rapid glacier acceleration (McMillan et al., 2014).

## 5.5 Frontal destabilisation of Stonebreen

In sum, we showed that the lower part of the southern lobe of Stonebreen has been subject to considerable losses in ice thickness. This could have made the tide-water calving glacier, which is grounded below sea level some 6 km inland from the 2014 front, more sensitive to either surface melt-water reaching its bed and/or warm ocean water increasing frontal ablation. Until 2014, such summer effects seem to have triggered a feedback mechanism where also winter speeds did not

fall back to pre-instability speeds, for instance due to basal strain heating or destruction of the subglacial drainage network and related increased basal water pressure, so that summer speed maxima of a specific year can reach higher speed levels than in the previous year. However, in early 2015 and 2016 the winter speeds were decreasing again as now observed thanks to the frequent standard satellite coverages

In contrast to the strong elevation losses between 1970 and 2014 in the lower part of Stonebreen (Fig. 3), the upper part of the glacier seems to have been quite stable in elevation over this period. This is in agreement with elevation changes found on Digerfonna on Edgeøya (Kääb, 2008). We have to leave open if these stable elevations at higher altitudes are a climatic signal from stable or even increased accumulation or from a dynamic imbalance of the glacier. However, the combination of strong elevation losses at the lower part (up to almost 2 m/yr as 40-yr average and up to ~20-30 m/yr between 2010-2012

and 2014) with stable (or perhaps even slightly increasing) elevations at higher sections shows that the glacier was not in a dynamic equilibrium before getting unstable.

Due to the lack of bathymetric data of sufficient quality, only a rough estimation of calving flux and sea-level contribution by the Stonebreen instability is possible. Assuming a water depth of about 25 m, an average front height above sea level of about 45 m (IDEM), a length of the calving front of around 6 km, an annual average speed of 1 200 m/yr, and pure sliding

gives a flux of ~0.5 km$^3$/yr through a flux gate close to the 2014 calving front. This value can be roughly partitioned into the advance of the calving front (on average ~ 350 m/yr for 2014-2015 corresponding to ~ 0.15 km$^3$/yr) and the actual frontal ablation (~ 0.35 km$^3$/yr). Again, these numbers are rough initial estimates that directly depend on the little known water depth in front of Stonebreen. For instance as some kind of upper bound, an average water depth of 50 m instead of the 25 m assumed would give a total sea level contribution of ~ 0.68 km$^3$/yr, and an average water depth of 15 m as some kind of

lower bound would give ~ 0.43 km$^3$/yr. These numbers show that the investigated instability currently plays an important role in the overall mass balance of Edgeøyjøkullen which was analysed by Nuth et al. (2010) using the 1970 NPI DEM and ICESat altimetry data over 2003-2007 period. Nuth et al. (2010) found for the entire Edgeøyjøkulen a volume change of 0.79 ± 0.15  km$^3$/yr, which corresponds to a geodetic mass balance of 0.58 ± 0.11 m/yr in water equivalent.

# 6 Conclusions

Though the process currently observed for Stonebreen does not seem a classical surge where a mass surplus in the accumulation area leads to increased speeds and travels down the glacier, the investigated instability could well have a similar effect when the section of increased speed extends upwards the glacier and is able to drain accumulated ice from the upper parts of the ice cap. Independent of its future development, however, the underlying cause of the instability seems to be an adjustment of the glacier front, which is grounded below sea level over some 6 km, to significant thickness losses during the recent decades. Influx of surface melt-water to the glacier base or of warm ocean water to the front could have triggered the instability and seasonally modulated its variation in surface speed over time. As similar decadal losses in ice thickness of slow-flowing calving fronts are also found elsewhere on Svalbard (e.g. over the northern lobe of Stonebreen or on the southwestern coast of Austfonna), processes like the one investigated here could happen also at these other locations or seem already to have started for the southeastern tip of Austfonna (sometimes called Basin-2) as seen in unpublished velocity maps based on Sentinel-1 data. The shape of the velocity field of Basin-2 resembles the one of the Stonebreen instability, but like on Basin-3 the speeds seem not to fall back on a winter level as low as for Stonebreen. However, a longer time-series might be needed for Basin-2 to draw more detailed conclusions. Also other glaciers in different regions, such as Pío XI in Patagonia (Mouginot and Rignot, 2015), presents similar features as Stonebreen, such as shallow bed below sea level at the terminus, large thickness changes, strong seasonal and annual variations, and large melt water production. Over the Canadian Arctic Van Wychen et al. (2016) introduced the concept of pulse-type glaciers. Over this kind of glaciers the velocity variability initiates in and propagates upglacier from the lowermost sections of the glacier near the terminus and is largely restricted to regions where the bed lies below sea level. Even if also for Stonebreen the instability initiated near the terminus, the velocity variability is now migrating upglacier more than the 6km inland from the 2014 front where the glaciers is believed to be grounded.

Through frequent monitoring based on repeat optical and SAR satellite data the future evolution of glaciers on Svalbard and elsewhere can be now recorded at high temporal sampling. A comprehensive temporal record of surface ice velocities is beneficial to glaciological studies, allowing more in deep understanding of glacier flow mechanisms. With the recent launch

of the Sentinel-1b satellite, regular SAR acquisitions are now available for many Arctic glaciers every 6 days. Radarsat-2 and ALOS PALSAR-2 can complement the Sentinel-1 data with SAR images at higher spatial resolution and lower wavelength, respectively, which might be more favourable in many cases. Also satellite optical images are currently available with a high temporal frequency (Landsat-8 data every 16 days and Sentinel-2 every 10 days), complementing radar data in the case of summer, cloud-free conditions. Future radar sensors, such as the planned Radarsat Constellation Mission, will even increase the temporal coverage up to 4 days.

## Acknowledgments

The research leading to these results received funding from the European Space Agency (ESA) within the Glacier_CCI project (code 400010177810IAM), the European Union Seventh Framework Program (FP7) under grant agreement No. 607052 (SEN3APP), the ERC grant agreement no. 320816, the Research Council of Norway through RASTAR (grant number 208013), and the Norwegian Space Centre as part of European Space Agency's PRODEX program (C4000106033).

ERS-1/2 SAR and ALOS PALSAR images provided by the European Space Agency, courtesy of AOPOL.4086. Sentinel-1 and Radarsat-2 Wide Ultra Fine images available from Copernicus. Radarsat-2 Wide data provided by NSC/KSAT under the Norwegian-Canadian Radarsat agreements 2007–2015. Landsat data available from the U.S. Geological Survey. ASTER Data from LPDAAC.

Ice surface velocity data from ALOS PALSAR and Sentinel-1 images are available at the databases of the Glacier_CCI (https://glaciers-cci.enveo.at/crdp2/index.html) and FP7 SEN3APP ( http://sen3app.fmi.fi) projects.

## Competing interests

A.K. is a member of the editorial board of the journal.

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

**Table 1. Sensors (MSS: Multispectral Scanner System, TM: Thematic Mapper, ETM+: Enhanced Thematic Mapper Plus, OLI: Operational Land Imager) and acquisition dates of the Landsat imagery used for the mapping of the glacier outlines.**

| Sensor | Date |
|---|---|
| Landsat 2 MSS | 16/07/1976 |
| Landsat 5 TM | 26/08/1994 |
| Landsat 7 ETM+ | 28/07/2011 |
| Landsat 8 OLI | 02/09/2013 |
| Landsat 8 OLI | 07/07/2014 |
| Landsat 8 OLI | 06/07/2015 |

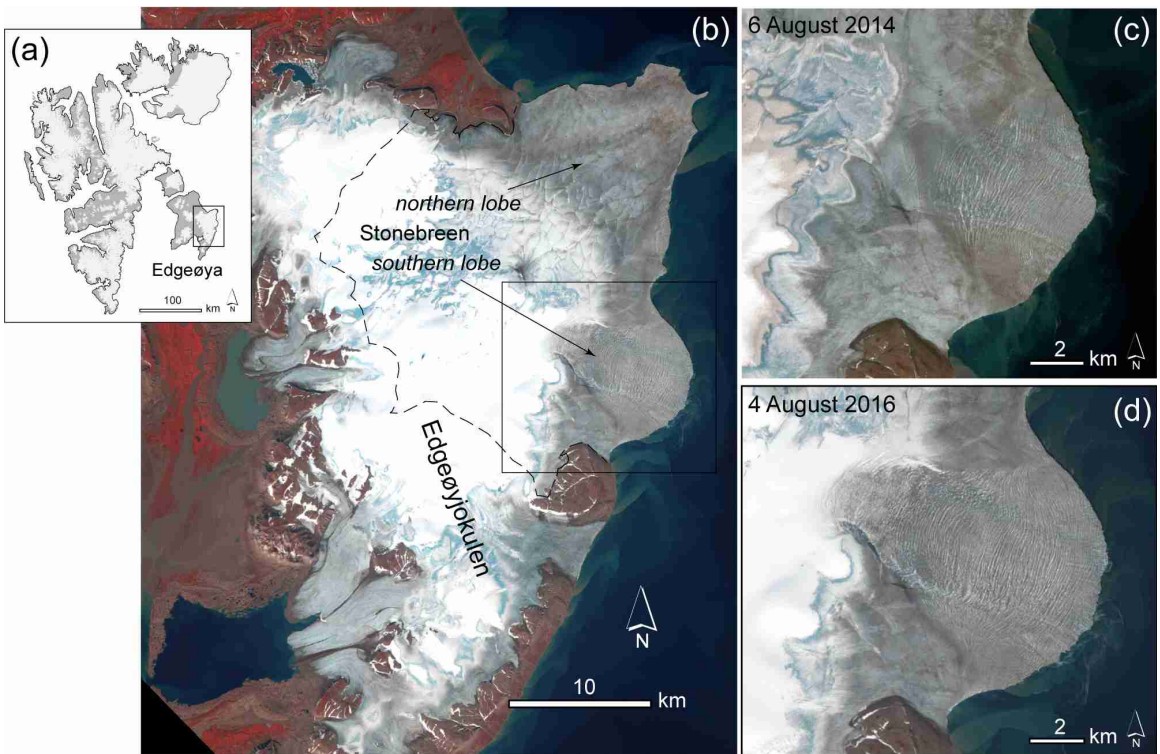

**Figure 1. Location of Edgeøya on Svalbard (a) and of Stonebreen on Edgeøya on a Landsat 8 image 04/08/2016 (b). (c) and (d) show the front of the southern lobe Stonebreen on two Landsat images of 06/08/2014 and 04/08/2016.**

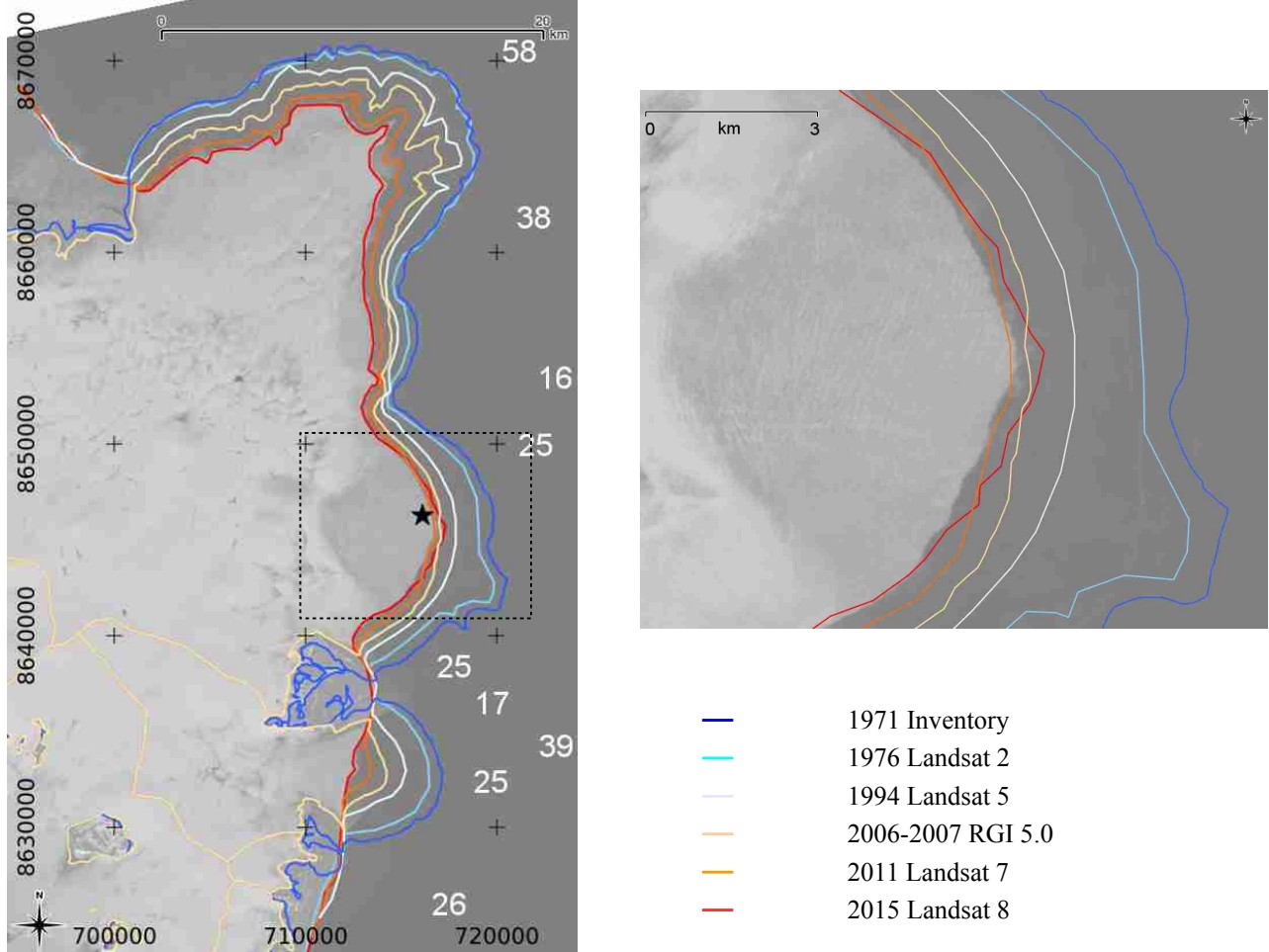

**Figure 2. Coastal outlines from vertical aerial photographs in 1971 (Nuth et al., 2013), Landsat imagery from 1976 to 2015 (this study), and the Randolph Glacier Inventory Version 5.0 (RGI, 2015) in 2006-2007 on a Landsat image of 14/07/2014. Easting and northing coordinates in meters are in the WGS 1984 UTM zone 33N. The star indicates the position of the profile on Stonebreen of Fig. 9. The light grey numbers indicate water depths from the Norwegian See Navigation Chart (2016). The inset shows a close up to the front of the southern lobe Stonebreen.**

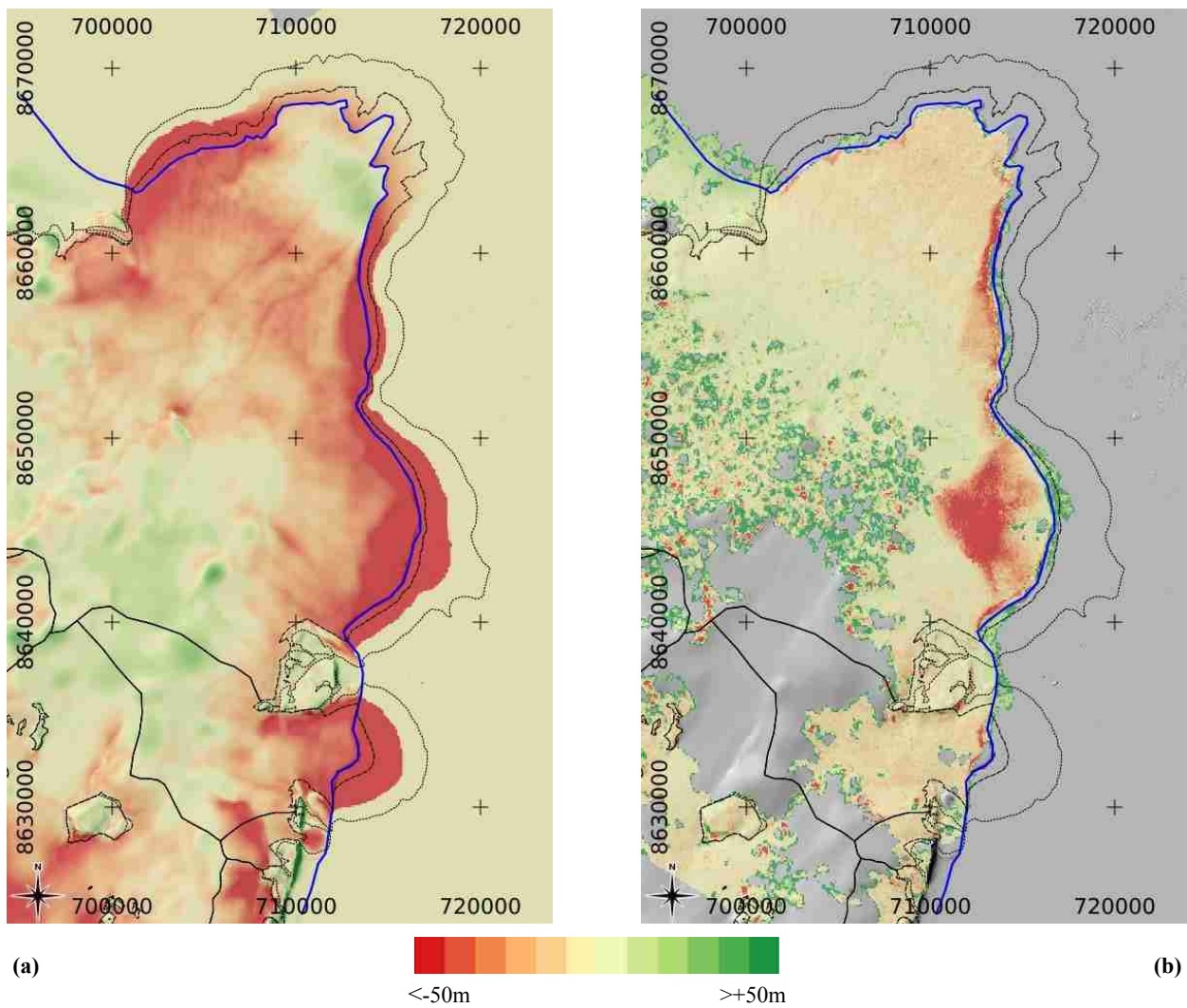

**Figure 3. Glacier elevation change between (a) IDEM and NPI DEM (time lapse 2010/2012 - 1970) and (b) ASTER DEM and IDEM (time laps 2014 - 2010/2012). Image background is a shaded relief of IDEM. The coastal outline from the Landsat imagery of 2011 (blue line) is shown along with glaciers inventories of 1971 (dotted line) and 2006-2007 (dashed line).**

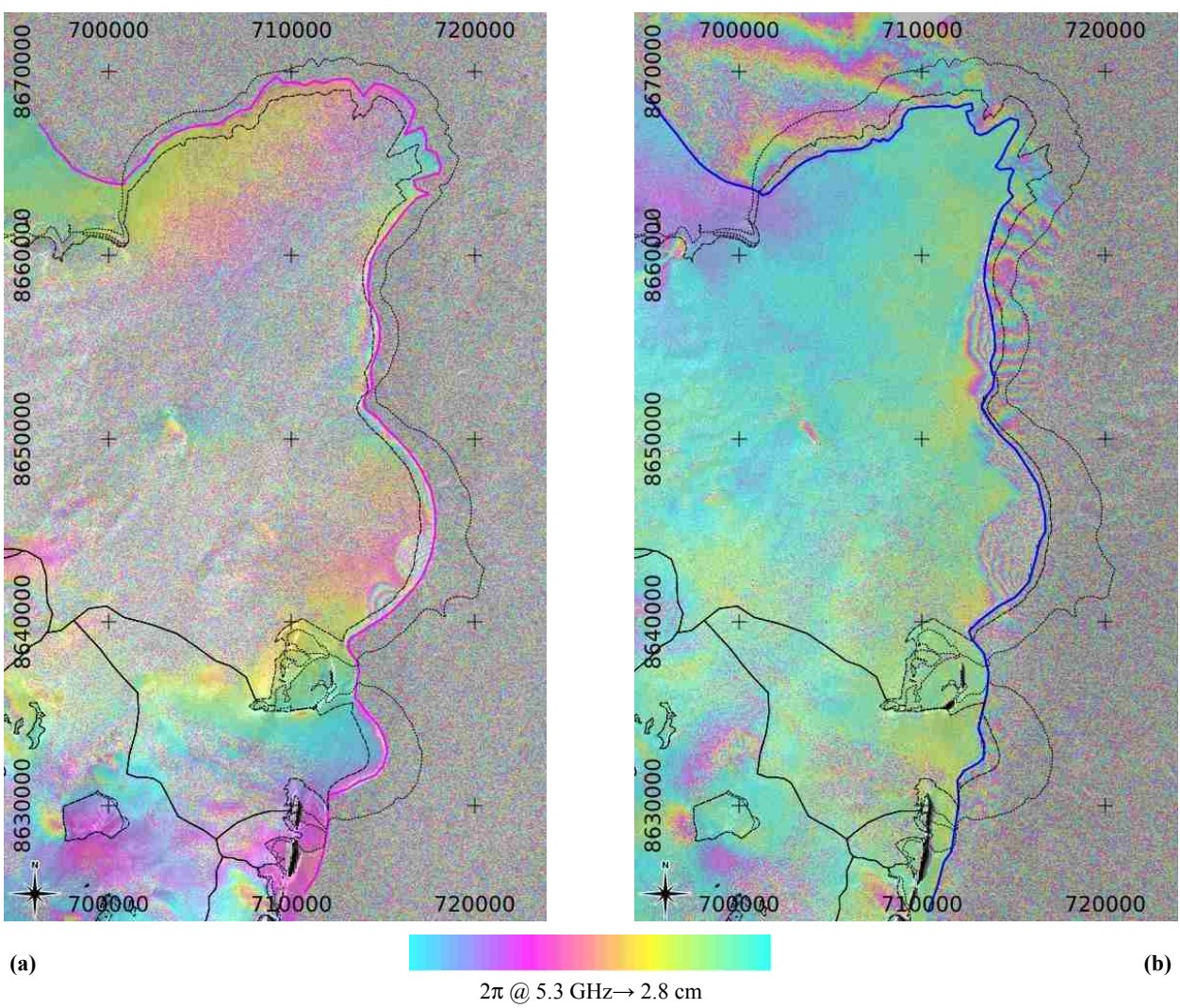

$2\pi$ @ 5.3 GHz→ 2.8 cm

**(a)**

**(b)**

**Figure 4. Differential SAR interferomgrams from (a) ERS-1 data of 02/01/1994 and 05/02/1994 and (b) ERS-2 data of 22/03/2011 and 25/03/2011. Image background is a backscattering intensity image of the master scene used for interferometry. The coastal outline on the year of the SAR images is shown along with glaciers inventories of 1971 (dotted line) and 2006-2007 (RGI 5.0, dashed line).**

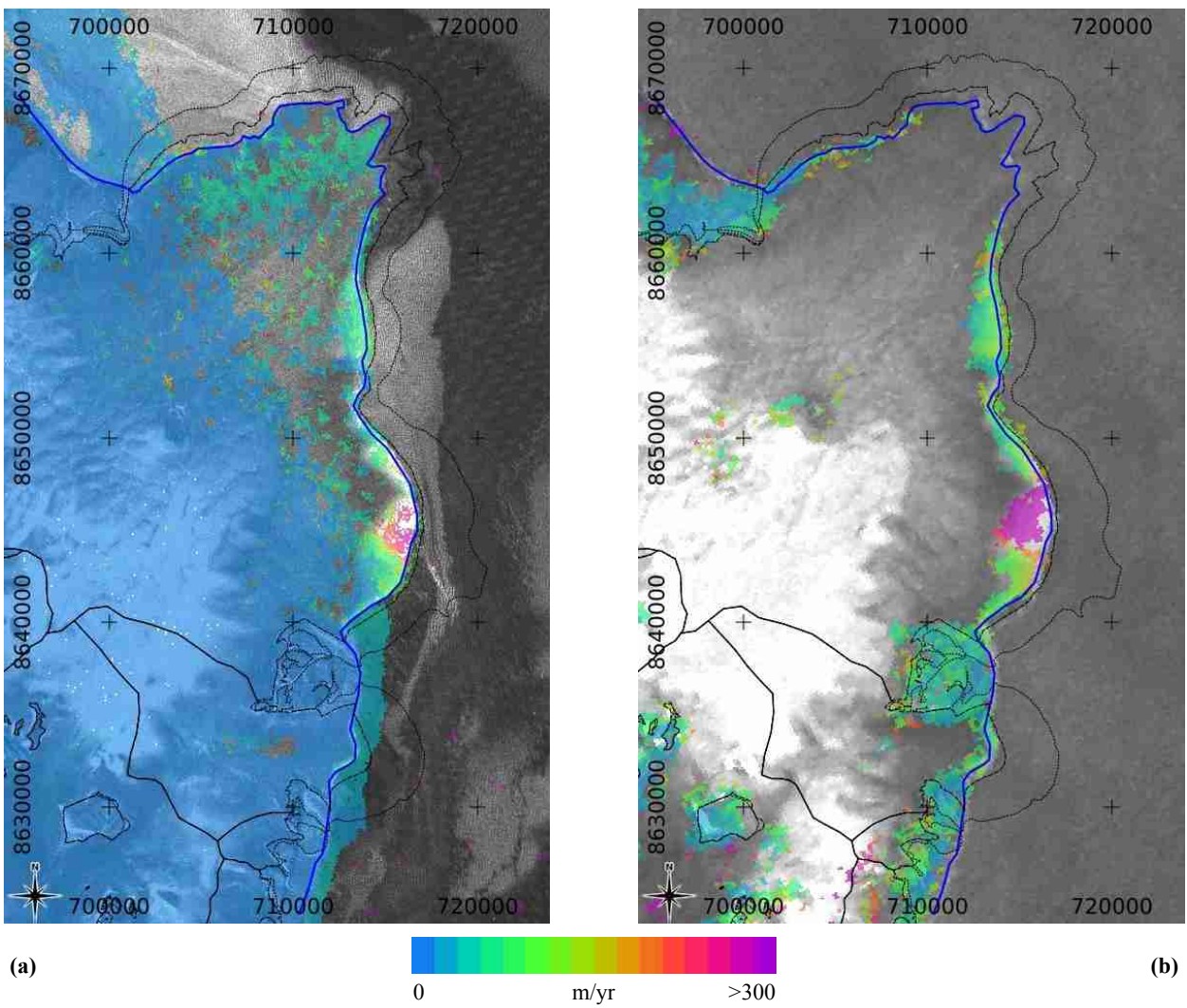

**(a)**

**(b)**

0                    m/yr                    >300

**Figure 5. Ice surface velocity maps from (a) ALOS PALSAR data of the winter 2010/2011 and (b) Radarsat-2 Wide data of 21/10/2011 and 08/12/2011. Image background is a backscattering intensity image of the master scene used for offset-tracking. The coastal outline in the summer of 2011 (continuous blue line) is shown along with glaciers inventories of 1971 (dotted line) and 2006-2007 (RGI 5.0, dashed line).**

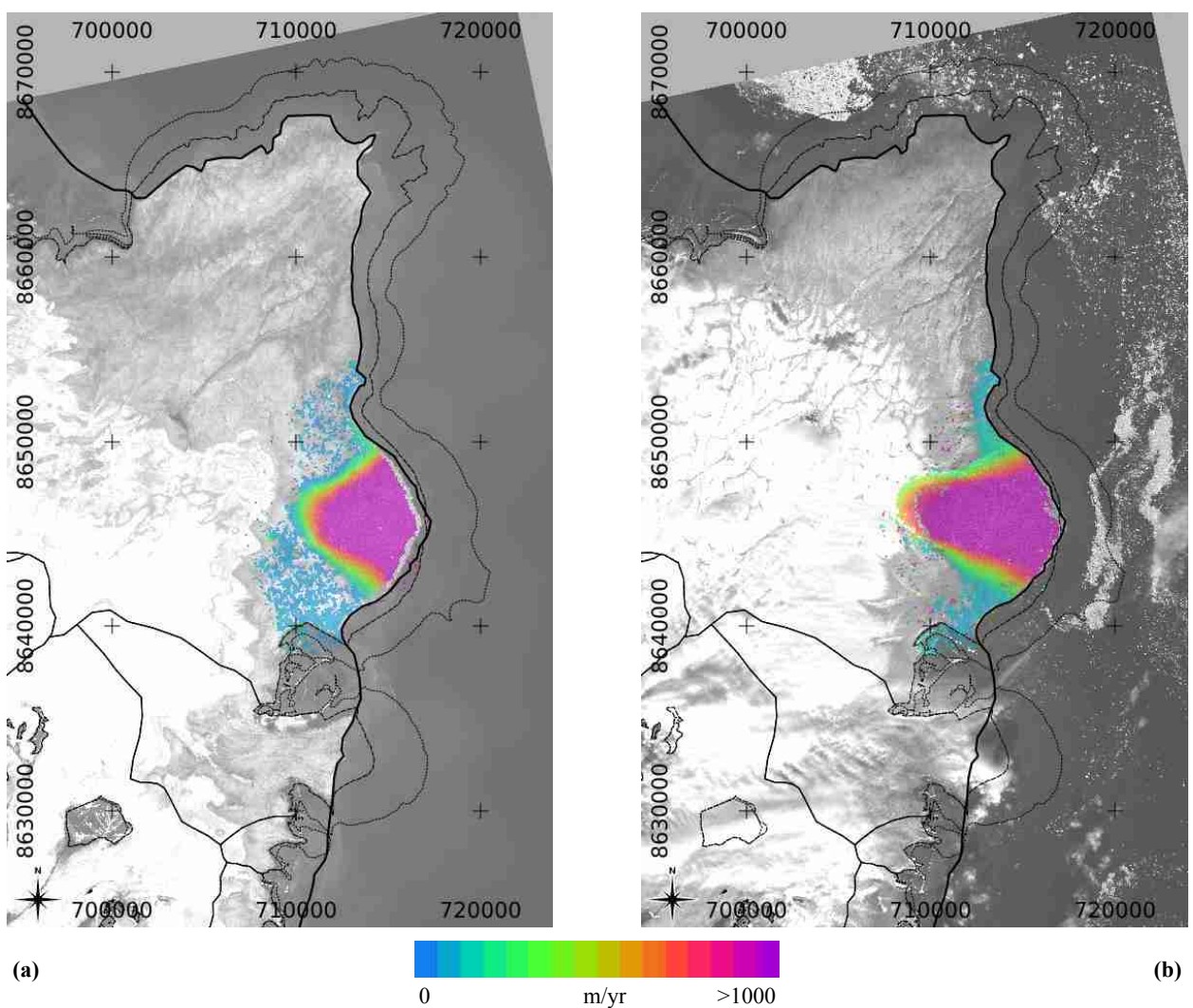

(a)

(b)

0   m/yr   >1000

**Figure 6. Ice surface velocity maps from Landsat 8 data of (a) 07/08/2014 and 25/08/2014 and (b) 19/08/2015 and 18/09/2015. Image background is from the respective first scene. The coastal outline in 2015 (continuous line) is shown along with glaciers inventories of 1971 (dotted line) and 2006-2007 (RGI 5.0, dashed line).**

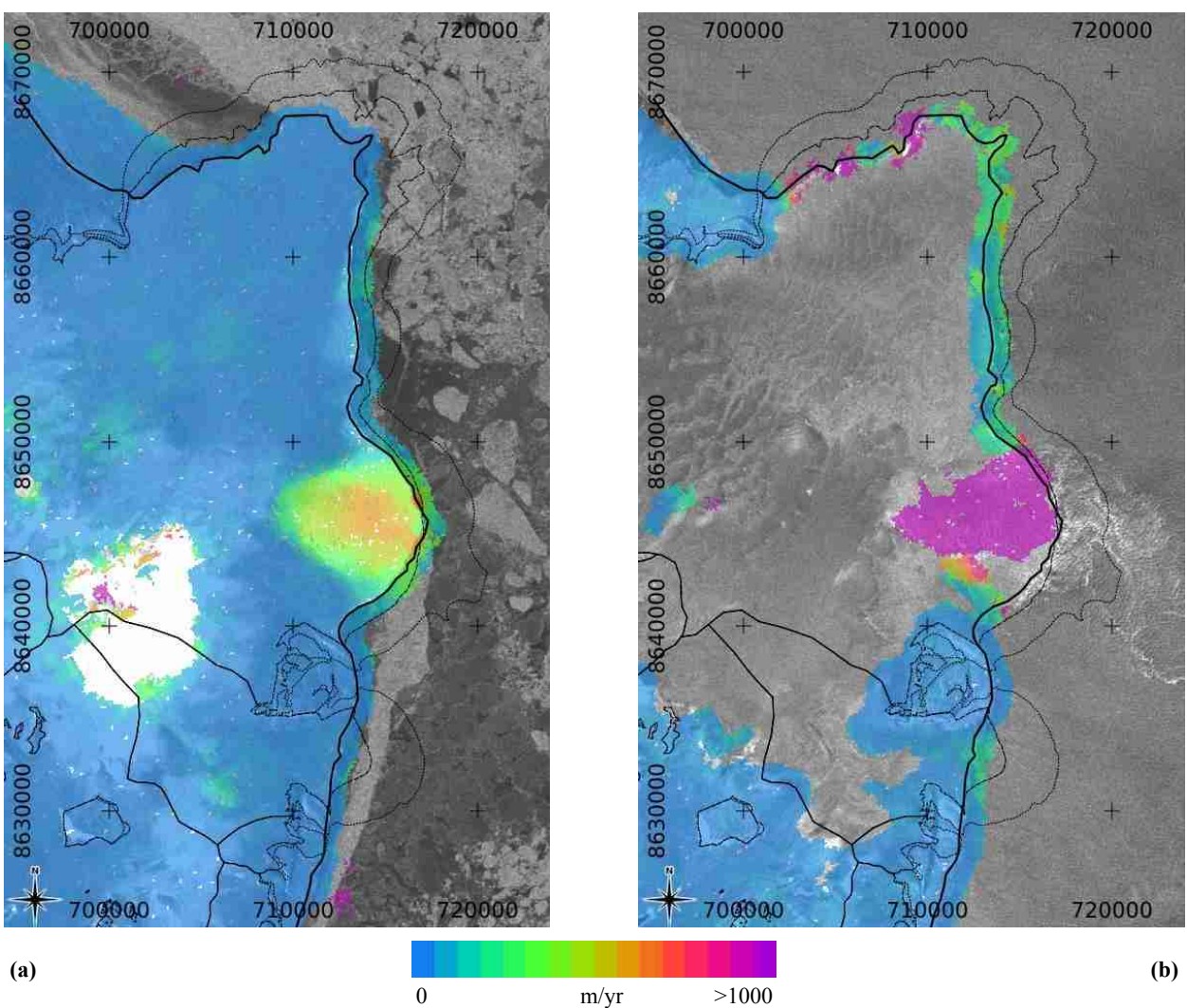

**(a)**

0         m/yr    >1000

**(b)**

**Figure 7. Ice surface velocity maps from (a) Sentinel-1 data of 21/01/2015 and 02/02/2015 and (b) Sentinel-1 data of 30/09/2015 and 12/10/2015. Image background is a backscattering intensity image of the respective first scene used for offset-tracking. The coastal outline in 2015 (continuous line) is shown along with glaciers inventories of 1971 (dotted line) and 2006-2007 (RGI 5.0, dashed line).**

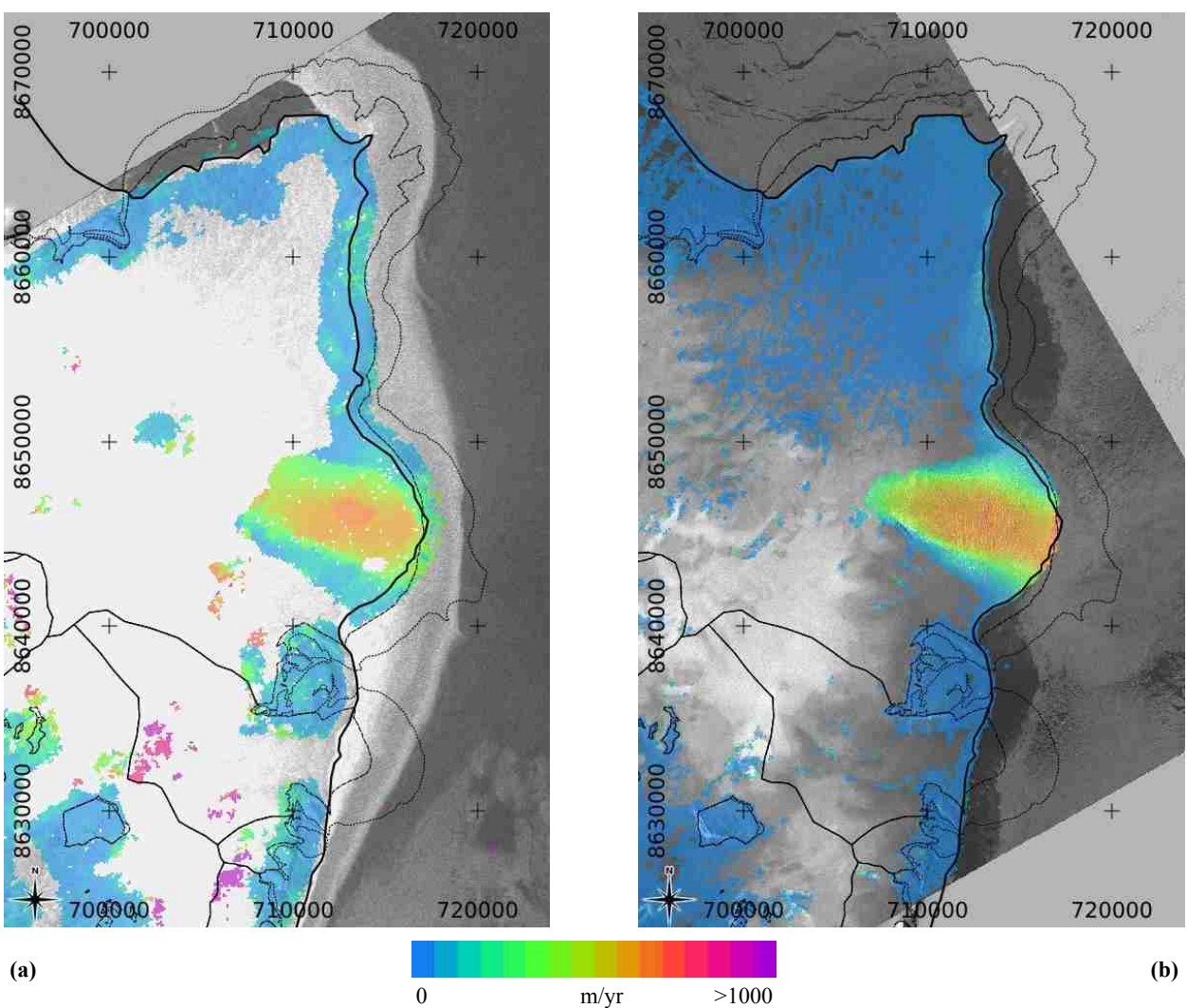

**Figure 8. Ice surface velocity maps from (a) Sentinel-1 data of 04/03/2016 and 16/03/2016 and (b) Radarsat-2 UWS data of 28/02/2016 and 23/03/2016. Image background is a backscattering intensity image of the master scene used for offset-tracking. The coastal outline in 2015 (continuous line) is shown along with glaciers inventories of 1971 (dotted line) and 2006-2007 (RGI 5.0, dashed line).**

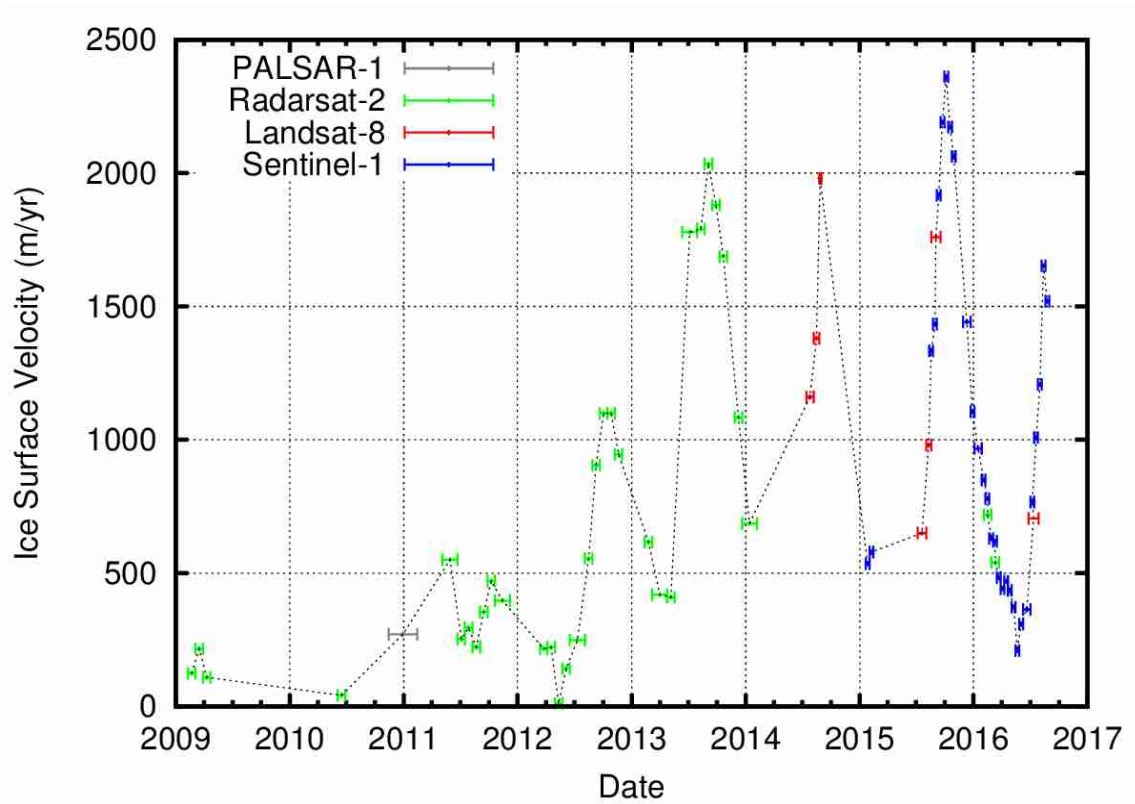

**Figure 9. Time-series of ice surface velocities close to the front of the southern lobe of Stonebreen (716080 E / 8646230 N). For position see Fig. 2.**

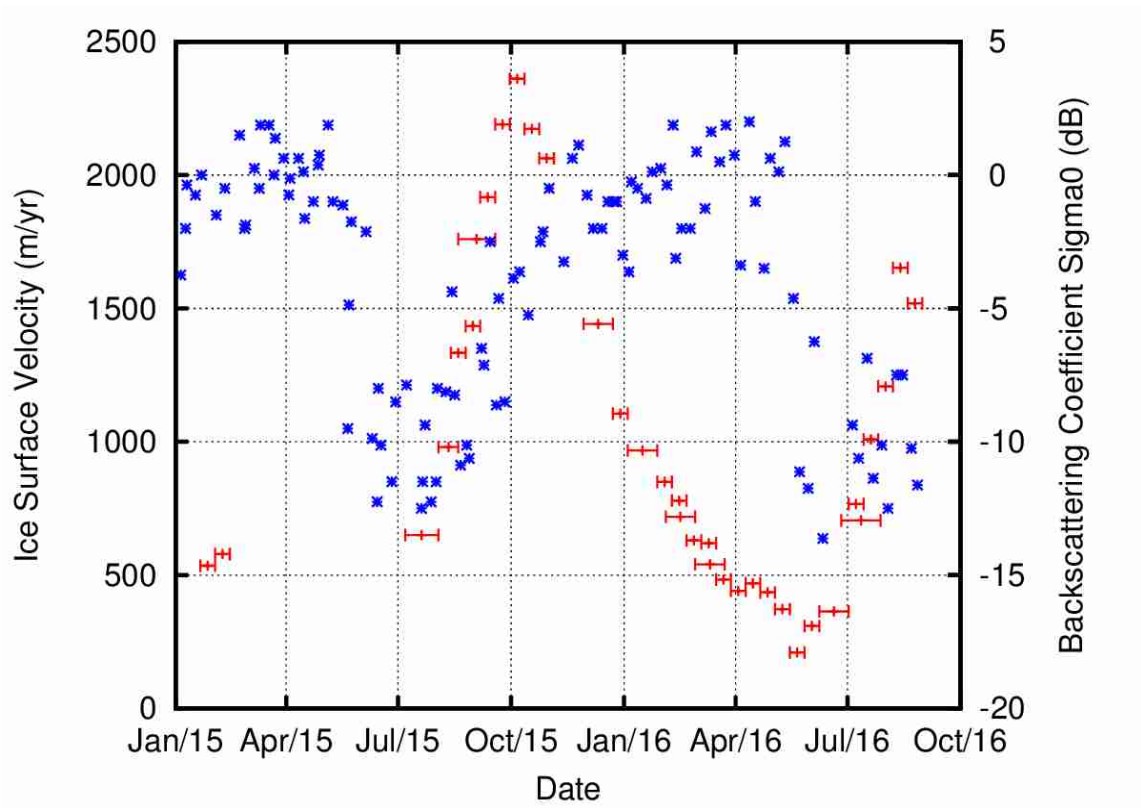

**Figure 10. Time-series of backscattering coefficient σ⁰ from Sentinel-1 EWS data (blue stars) along with the time-series of ice surface velocities from Sentinel-1 IWS data (red crosses) close to the front of the southern lobe of Stonebreen (716080 E / 8646230 N). For position see Fig. 2.**