# Peer review of "Frontal destabilisation of Stonebreen, Edgeøya, Svalbard"

_The Cryosphere, 2016_

## Referee Comment (RC1) · Anonymous Referee #1 · 28 Oct 2016

Strozzi et al. presents their study about the dynamic changes of Stonebreen located in the southeast of the Svalbard archipelago. Using remote sensing (mostly SAR), they have reconstructed the surface ice velocity since 1994. The glacier shows a strong acceleration from 1994 to 2016 superimposed with very strong seasonal variations (<0.5 km/yr in winter to >2 km/yr at the end of summer). The authors discuss the different causes for the glacier destabilization. They conclude that surface melt-water and/or warm ocean water could be the cause of such changes.

I do not see any issues with the processing and analysis of the different remote sensing data sets (surface ice velocity changes). The results shown here are solid and should be published.

On the other hand, the discussion about the potential causes for the glacier acceleration is not well supported due to the lack of other external data (bathymetry, ice thickness, ocean water temperature). The authors are seeing strong seasonal variations of the ice speed and mention the frontal ablation as a cause for the observed changes. To better prove this interaction, I think it would be interesting to show the seasonal position of the terminus corresponding to Figure 9 and Figure 10 and see if they are linked to seasonal speed changes. Although the authors rule out ice thickness changes as the cause of the recent speed fluctuations, I still believe that the combination of higher input of melt-water and ice thickness reduction could have triggered this surge-type behavior.

The glacier shows similar behavior that other "surge-type" glacier such as Pío XI in Patagonia (see Fig. 2c in Mouginot and Rignot 2015), which presents similar features such as shallow bed below sea level at the terminus, large thickness changes, strong seasonal and annual variations, and large melt water production. I believe a comparison with other glaciers in the region or elsewhere would be interesting. In other words, is the behavior of Stonebreen glacier unique, and if yes, in what sense?

Below are the minor comments on the document:

L10 don't -> do not

L1,2 : unclear. If at steady state, calving fluxes are always the same order of magnitude than surface mass balance. The authors probably meant that mass fluctuation in Svalbard is similarly controlled by both dynamic and SMB changes.

The authors mention ERS data with 3-day repeat cycle are not suitable for speckle tracking. I wonder if the authors looked at longer repeats (6 to 36 days). I know in Greenland such pairs are sometimes available, is it the case over Svalbard ?

The authors did not mention ionosphere noise in your ALOS error estimation. It is probably very small (not visible in Fig. 5a) or is it a source of error here ?

I think a reference would be needed for the computation of speed changes from increase in slope.

L21: "Total contribution to sea level..." sentence is not clear as described here. If the authors look at the calving flux, they have to compare to the surface mass balance. they could assume that surface mass balance was equal to zero (no discharge and glacier in balance), but if they do so, they should state it. In conclusion, more details on contribution to sea level needed here.

Page 12 or 13

I think the derived data sets should be made available to the scientific community. A sentence in the conclusion or acknowledgments where to find them would be great.

Table 2 could be added as supplemental material. I see that Landsat-8 pairs are not a factor of 16 days (nominal repeat cycle), which means that the authors used different path/row to compute ice speed. It is a potential large source of error due to topographic effects (even with the orthorectification from USGS). I would recommend using only identical orbits as done for SAR sensors.

Figures should be vector graphic rather than raster.

Fig. 1

If Stonebreen glacier is the glacier shown in Fig.1c-d, I think the label Stonebreen in Fig.1b should be placed differently. Perhaps an arrow pointing to glacier would do.

Fig. 2

The background is not contrasted enough, which makes the map difficult to read

Fig. 5-8

These figures could be combined in one figure.

Fig. 10

Although obvious, blue and red dots should be explained in caption. Corresponding terminus position would be a must.

---

## Referee Comment (RC2) · Anonymous Referee #2 · 31 Oct 2016

General Description:

Strozzi et al., [2016] use optical and SAR datasets to observe the dynamic evolution of the Stonebreen Glacier of Edgeøyjøkulen Ice Cap (Svalbard) in 1994, and more frequently from 2011-2016. They combine the observed pattern of velocity fluctuation with glacier geometry evolution (determined from DEM differencing) and terminus position change, in order to speculate at the mechanisms causing the observed fluctuations in surface motion. A secondary stated goal is to evaluate the potential of frequent standard coverage acquisitions from recent earth observation missions (such as Landsat-8 and Sentinel-1) in order to analyze temporal variability in ice motion. The authors draw upon well used methods, and although there is little novel with regard to the methodology, they are nevertheless suitable for this type of work and reasonable uncertainty levels are provided. For the most part, the paper is well written and easy to follow,

although in some places small typos need to be corrected and some modified work choice would help improve clarity. The tables are generally well done and complete, although some of the figures may be combined to improve clarity.

At present, the discussion section of the paper is a bit too brief. I suggest that the section begin with a short paragraph which outlines how the velocity variability observed here differs significantly from glacier surging (which is observed in other basins of Edgeøyjøkulen Ice Cap), once the distinction from surging is made clear, then the other mechanisms that may be causing the variability in ice motion and the reasons for and against each of these mechanisms from the observations, can be described. I suggest that the authors also look at the "pulse" mechanism described by Van Wychen et al., [2016] for the Canadian Arctic as another mechanism that may be inducing fluctuations in ice motion. Finally, given that a major goal of this work is to assess the importance of frequent standard coverages of earth observation data for glacier velocity monitoring, there needs to a portion of the discussion devoted to this topic and more than a single sentence regarding this topic in the concluding remarks. Despite these comments, the authors now provide a much more comprehensive record of ice velocities for Stonebreen than was previously available and the dynamic behaviour observed here may apply to other glaciers in Svalbard (and other Arctic regions). I have provided a number of points below for the authors to address.

**Specific Comments**

Minor Changes

PAGE 1

L6: Please provide a reference for the warming trend observed since the 1990s.

L7: "ice mass loss" -> "mass loss"

L11: "glacier's" -> "glacier"

L12: suggest changing "speed increases" to "velocity increases"

L13: Please provide references.

L14-15: "from 1971 until 2011 followed since 2012 by a strong increase in ice surface velocity along with a decrease of volume and an advance in frontal extension" -> "from 1971 until 2011, followed by (since 2012), a strong increase in ice surface velocity along with a decrease of volume and frontal extension".

PAGE 2

L3: "The total calving flux of Svalbard is dominated by a few large and fast-flowing glaciers" please provide a reference.

L4: "So, far" -> "So far,"

L5: "A few glaciers" -> "A few glaciers," add comma

L9: "overdeepenings in the glacier bad" -> "overdeepenings of the glacier bed"

L10: "reduced buttressing" and "changes in the back-stressing sea ice cover in front of the glaciers" are these differing mechanisms? If so, please clarify the distinction.

L11-12: Please provide a reference or example to back up the statement.

L14: "of Svalbard" -> "of the ice masses of the Svalbard Archipelago"

L22: "seem possible" -> "seems possible"

L24: "data a" -> "data, a" (add comma)

PAGE 3:

L4: Please provide a lat/long coordinate for Stonebreen.

L6-7: "new missions" -> "new earth observation missions" such as Sentinel-1 (SAR) and Landsat 8 (optical) to detect..."

L10: "5,073 km2" -> "5,073 km2" change to superscript.

L11: "The eastern side of Edgeøya is covered by the Edgeøyjøkulen Ice Cap, which had an area of 1365 km2 in 1985"... (also note the superscript) -> please modify text.

L12-13: "is among the least well" -> "are among the least well", also please add the (Dowdeswell and Bamber, 1995) reference to this statement.

L19: "extension" -> "area"

L20: Mark the northern lobe of Stonebreen with an "\*" and the southern lobe of Stonebreen with an "#" on Figure 1 to make it clear to the reader exactly where you are referring to. Also suggest making all the glaciers previously identified as surge-type (e.g. from Liestøl, 1993 and Dowdeswell and Bamber, 1995) with an "\*".

PAGE 4:

L7-11: Please provide an indication of the relatively uncertainty between glacier delineations versus pixel size between sensors. Have all the images been georeferenced to a common image?

L15: Please provide an uncertainty value for the NPI DEM.

L23: "Digerfonna Kääb (2008)" -> "Digerfonna Ice Cap, Kääb (2008)"

PAGE 6:

L1-9: Please provide a description of the window sizes used for the SAR offset matching algorithm.

L12: "Landscape 8" -> "Landsat 8"

L13: "For good visual contrast such as given for our study site and data due to the crevassed and snow-free glacier" -> "For areas of good visual contrast, such as those in our study site due to crevassed and snow-free glacier surfaces, displacement accuracies..." also please provide a reference for these values.

L1-16: For both the SAR offset tracking and the optical feature tracking, please describe how mis-matches or blunders are removed from the dataset. Was this completed manually? Was the strength of the cross-correlation value used to flag poor matches? Please describe in more detail.

L20: "on a Landsat image of 14/07/2014" this can be removed as it appears in the figure caption.

PAGE 7:

L2: "NPI DEM of 1990" -> "NPI DEM of 1970"

L3: "From 1990" -> Please check here and throughout the manuscript and figures. You note that the NPI DEM is derived from aerial photography in 1970 (Page 4, L16), however at other times in the manuscript (see section 4.2 you describe it as being from 1990). Please correct.

L2-4: This section is somewhat awkwardly phased and could be clarified and would benefit from further description. Please describe more fully what "height losses of up to 150 m over current sea level and up to 100 m over current ice" really means. Suggest changing "current" to the last year when DEM data is available. It is noted that height changes of 100-150 m are observed, however the Figure scaling only shows elevation changes +/- 50 m, please adjust the scale bar so that the description in texts describes what can fully be seen in the figure.

L13: "The velocity is lower towards south" -> "The velocity is lower towards the south"

L14: "The northern sector is decorrelated, i.e. flowing faster" -> Yes, this can cause decorrelation, but what about change in the glacier surface that could cause decorrelation? Provide further evidence why you attribute it to faster flow speeds rather than changes in surface characteristics.

L17: "are indicating" -> "indicate"

L19: "in summer of 2014 (a) respectively 2015 (b)" -> awkwardly phrased, should be

re-written.

L20: "with a migration of the front of increased speeds towards inland" -> "due to an inland migration of a front of elevated velocities"

L20: "1'500" -> "1,500" or just "1 500"

L24: "12-days" -> "12-day"

PAGE 8:

L5-6: "dynamically active sector is increasing again inland" -> does this mean that the dynamically active sector is again migrating inland?

L13: "2'500" -> "2,500" or "2 500"

L14: "The different SAR and optical satellite sensors complement each other very well". In principal I agree with this statement, however I would like to see the authors develop this idea further, especially because evaluating the potential of frequent standard acquisitions is stated as a secondary goal of this paper (Page 3, lines 6-8).

To further illustrate this statement, I suggest creating a timeline figure that shows all the image acquisition broken down (colour coded) by sensor for the period from  $\sim$ 2010-2016. For an example, see bottom panel of Figure 2 in Burgess et al., [2012] of how this can be accomplished. I recognize that this information is available in Table 2, however the visual timeline would show the ready more easily how much of the time during the study period that the site was under observation, and further highlight the point that frequent observations improve our understanding of the temporal evolution of glacier velocities. This newly created figure may have the potential to replace Table 2, or at least move that table to supplementary materials.

L20: "increase in slope" -> "increase in surface slope"

L22: "increment" -> "increase"

PAGE 9:

L7-8: "The increased elevation loss towards the front lead to an increase" -> "The elevation loss at the glacier front between the NPI DEM and the IDEM led to an increase in surface slope of  $\sim\!2^\circ\!\ldots$ "

L18-20: This sentence is somewhat awkward to me and should be rewritten to improve clarity.

PAGE 10

L10-12: The method for extracting and comparing backscatter intensity needs to be more fully described, and this should be provided in the methods section. Have all the backscatter intensity values been corrected to sigma nought values to account for various incident angles to enable comparison from different acquisitions and incidence angles? Or can this be neglected because all of the images are interferometric pairs with the same viewing geometry? This is not clearly described in text and should be. In addition to only using the backscatter values to determine melt rates, is it possible to use nearby meteorological station data or NCEP reanalysis to strengthen your claims?

PAGE 11

L13-17: This portion of the paragraph is somewhat unclear and could be tidied to improve clarity.

PAGE 12:

L1-5: These sentences can be modified to improve clarity.

L10: suggest changing "(surge-type?) instability" to just "instability"

L15-L19: Comparisons with unpublished data for Basin-2 should be presented at the end of the discussion section rather than being introduced within the conclusions.

L21: "at high temporal sampling", suggest quantifying this remark. How often will

Svalbard be covered with standard acquisitions in the future? Every 12 days going forward? Or does it change seasonally? Provide a bit more information here.

PAGE 13:

L4: "The Reearch Council of Norway" -> "The Research Council of Norway"

PAGES 13-16:

Check all references, in some cases DOI numbers are missing.

Substantive Comments:

Methods/Results Sections

Please further discuss the comparison of Sentinel-1 backscatter values over the melt season. Currently, this topic only appears in the discussion section, but should also be described in the methods and results sections.

**Discussion Section**

I would like to see the discussion section begin with a brief description of why the observed velocity pattern does/does not conform to traditional surge theory, which then narrows down to introduce the alternate processes provided by the authors that could explain the velocity variability.

One potential mechanism that is not described by the authors, but may be relevant, is "pulsing" which has been observed in other Arctic regions (see Van Wychen et al., 2016). This mechanism involves geometry changes, glacier advance and glacier speed-up, and the authors may want to include this as another potential mechanism in their discussion.

Given that the stated secondary goal of the paper is to evaluate the potential of frequent standard coverages of earth observation data to analyse glacier dynamics there needs to be a portion of the discussion section devoted to this topic. Currently, the discussion section does not provide any reason why the authors believe that frequent standard coverages are beneficial beyond a very brief statement. Although this may seem somewhat obvious, it needs to be discussed fully if the authors intend on it being a major outcome of the paper.

**Conclusion Section**

Again, given that the secondary goal of the paper is to show how beneficial it is to have frequent coverage of earth observation datasets for glacier monitoring, the conclusions should devote more than one sentence to this topic. Specifically, the authors should note that with the data they had available, that they were able to monitor the dynamic evolution of this glacier nearly continuously from  $\sim$ 2011-2016, and that this likely would not have been possible in the recent past. The authors may also want to speculate as to how recently launched (Sentinel 1b) or future sensor (Radarsat Constellation Mission) will even further increase the amount of data available for this type of monitoring.

**FIGURES:**

Figure 1: needs to be modified and clearly indicate that the ice cap is named "Edgeøyjøkulen" and that "Stonebreen" is a glacier basin within the Edgeøyjøkulen Ice Cap. Suggest adding the glacier basin delineations from the GLIMS Randolph Glacier Inventory and provide an arrow to the Stonebreen Glacier Basin. Suggest also adding a notation, such as "\*" to the basins that have previously been identified as "surge type" in the literature. Increase the size of the north arrows as well as the font of the scale bars (particularly on (b), (c), (d)) to improve readability.

Figure 2: Please provide the background image as a panchromatic image rather than a multi-spectral image, right now the image appears washed out and for clarity would appear better as a grayscale background image. Suggest changing the colour scheme of the glacier outlines and use a graded colour scheme (blue to red with time) rather than a mixture of colour and gray outlines. Please add a scale bar to this figure as it will aid the reader to determine the scale of terminus position change along the calving

front. Suggest adding an inset map to show the frontal advance of the southern lobe of Stonebreen between 2011 and 2015, at the current scale it is difficult to see.

Figure 4: It would be more beneficial if these figures were projected to velocities rather than presented as interferomgrams. This would enable comparison of glacier velocities shown in Figures 5-8 and may be beneficial to readers that are not familiar with interpreting interferometric fringes.

Figures 4-8: The authors should consider combining Figures 4-8 into a single figure with multiple panes and with a common glacier velocities scaled colour bar. By combining these figures together it would help the reader understand the dynamic evolution of the glacier more clearly. Also note, in figures 5-8, that the glacier velocity colour bar and the velocities provided on the map are fully saturated at the high end of the velocity bar, please consider increasing the colour bar scale to provide more distinction between velocity bands.

Figure 10: It may be beneficial to add a trend line for both data series which shows that RADAR backscatter values decrease as glacier velocities increase (albeit with some temporal lag) to indicate that melt may be modulating ice flow. Also, the figure caption needs to be more description, e.g. it needs to say that the blue marking indicate backscatter values and that red markings indicate ice surface velocities.

**REFERENCES:**

Burgess, E.W., Forster, R.R., Larsen, C.F., Braun, M. [2012], Surge Dynamics on Bering Glacier, Alaska, in 2008-2011. The Cryosphere, 6, 1251-1262, doi: 10.5194/tc-6-1251-2012.

Van Wychen, W., Davis, J., Burgess, D.O., Copland, L., Gray, L., Sharp, M., and Mortimer, C. [2016], Characterizing interannual variability of glacier dynamics and dynamic discharge (1999-2015) for the ice masses of Ellesmere and Axel Heiberg Islands, Nunavut, Canada. Journal of Geophysical Research: Earth Surface, 121, doi:

10.1002/2013JF003839.

---

## Author Comment (AC1) · 9 Jan 2017

**Response to Anonymous Referee #1**

Our response to the Referee #1 comments are given below and appropriate changes to the paper have been included in the new version of the manuscript. A "Track Changes" version and a "clean" version of the revised manuscript were prepared.

*Strozzi et al. presents their study about the dynamic changes of Stonebreen located in the southeast of the Svalbard archipelago. Using remote sensing (mostly SAR), they have reconstructed the surface ice velocity since 1994. The glacier shows a strong acceleration from 1994 to 2016 superimposed with very strong seasonal variations (<0.5 km/yr in winter to >2 km/yr at the end of summer). The authors discuss the different causes for the glacier destabilization. They conclude that surface melt-water and/or warm ocean water could be the cause of such changes.*

*I do not see any issues with the processing and analysis of the different remote sensing data sets (surface ice velocity changes). The results shown here are solid and should be published.*

Thank you very much for this encouraging remark and the constructive review given below that helped us to revise the paper.

*On the other hand, the discussion about the potential causes for the glacier acceleration is not well supported due to the lack of other external data (bathymetry, ice thickness, ocean water temperature).*

Yes, we fully agree. Unfortunately, these other external data are not available for our analysis, and, to our best knowledge, they do not actually exist at all. By necessity our discussion of potential reasons becomes thus open. Still, we believe it is important to publish information about these processes and make them thus more widely known.

*The authors are seeing strong seasonal variations of the ice speed and mention the frontal ablation as a cause for the observed changes. To better prove this interaction, I think it would be interesting to show the seasonal position of the terminus corresponding to Figure 9 and Figure 10 and see if they are linked to seasonal speed changes.*

In order to map the position of the terminus in correspondence to Fig. 9 and Fig. 10 we have to use Radarsat-2 and Sentinel-1 data, the only data set that are available a high temporal sampling throughout the year. However, the spatial resolution of Radarsat-2 and Sentinel-1 is not very high (on the order of 10 to 20 m) and the peculiar properties of the SAR sensors (side-locking, radar speckle) make it even more difficult to map the glacier's frontal position. By applying a certain multi-looking to the Sentinel-1 SAR data (e.g. 20 in slant-range and 4 in azimuth) we can reduce to a certain level the speckle, but at the end the terminus position cannot be identified with an accuracy better than 100 m. Considering also that calving happens along different positions of the front at different times, the resulting plot of the terminus position determined with Radarsat-2 in 2013 and 2014 at 24 days temporal sampling and with Sentinel-1 in 2015 and 2016 at 12 days temporal sampling is very noisy and not helpful for our discussion.

*Although the authors rule out ice thickness changes as the cause of the recent speed fluctuations, I still believe that the combination of higher input of melt-water and ice thickness reduction could have triggered this surge-type behavior.*

As written at lines 1 and 2 of page 3, we are actually not ruling out this cause but state that it could be a cause: "This suggests that reduction of ice thickness is not a result of the increase in flow and discharge to the ocean, but rather an independent process or cause." We now expanded and clarified

this sentence.

*The glacier shows similar behavior that other "surge-type" glacier such as Pío XI in Patagonia (see Fig. 2c in Mouginot and Rignot 2015), which presents similar features such as shallow bed below sea level at the terminus, large thickness changes, strong seasonal and annual variations, and large melt water production. I believe a comparison with other glaciers in the region or elsewhere would be interesting. In other words, is the behavior of Stonebreen glacier unique, and if yes, in what sense?*

According to the extensive comments of Referee #2, we revised the discussion to put the frontal destabilization of Stonebreen more in the general context of surges over Svalbard and elsewhere. We refer therefore to the response to Referee #2 for more information on this topic.

*Below are the minor comments on the document:*

*Page 1*
*L10 don't -> do not*
Done.

*Page 2*
*L1,2 : unclear. If at steady state, calving fluxes are always the same order of magnitude than surface mass balance. The authors probably meant that mass fluctuation in Svalbard is similarly controlled by both dynamic and SMB changes.*
Agreed, this sentence was reformulated. "For Svalbard, calving fluxes are assumed to be on a similar order of magnitude than the surface mass balance, making glacier dynamics an important factor of glacier's mass turnover and change."

*Page 5*
*The authors mention ERS data with 3-day repeat cycle are not suitable for speckle tracking. I wonder if the authors looked at longer repeats (6 to 36 days). I know in Greenland such pairs are sometimes available, is it the case over Svalbard?*
We tracked ERS and ENVISAT data over 35 days over Nordaustlandet in the past, but the results were of low quality and limited to the very crevassed fronts of the large active glaciers. The southern lobe of Stonebreen was not crevassed during the 1990's, though.

*Page 6*
*The authors did not mention ionosphere noise in your ALOS error estimation. It is probably very small (not visible in Fig. 5a) or is it a source of error here?*
Indeed ionospheric artifacts are not a source of visible error in Fig. 5a.

*Page 7*
*I think a reference would be needed for the computation of speed changes from increase in slope.*
Done.

*Page 11*
*L21: "Total contribution to sea level..." sentence is not clear as described here. If the authors look at the calving flux, they have to compare to the surface mass balance. They could assume that surface mass balance was equal to zero (no discharge and glacier in balance), but if they do so, they should state it. In conclusion, more details on contribution to sea level needed here.*
Agreed, "This total sea level contribution ..." was modified to "This value ...".

*Page 12 or 13*

*I think the derived data sets should be made available to the scientific community. A sentence in the conclusion or acknowledgments where to find them would be great.*

Agreed, we have now included in the Acknowledgments the server database addresses of the ESA Glacier_CCI and FP7 SEN3APP projects, where ALOS PALSAR and Sentinel-1 ice surface velocity data are available. Radarsat-2 Wide data can eventually be made available at a later point as the University of Oslo currently has no facilities to provide that kind of service.

*Table 2 could be added as supplemental material.*

Agreed, this could be added as supplemental material.

*I see that Landsat-8 pairs are not a factor of 16 days (nominal repeat cycle), which means that the authors used different path/row to compute ice speed. It is a potential large source of error due to topographic effects (even with the orthorectification from USGS). I would recommend using only identical orbits as done for SAR sensors.*

Indeed, Landsat-8 image pairs are not always the optimal factor of 16 days nominal repeat cycle, this depended on the availability of cloud-free summer images. However, we think that topographic effects are marginal. We have now included this aspect in the revised version of the manuscript and extended the last paragraph of Section 3.3 substantially in this respect

*Figures should be vector graphic rather than raster.*

Figures were prepared in a GIS using a combination of vector and raster layers, then exported to be included in the manuscript.

*Fig. 1*
*If Stonebreen glacier is the glacier shown in Fig.1c-d, I think the label Stonebreen in Fig.1b should be placed differently. Perhaps an arrow pointing to glacier would do.*

The following changes were included in the revised version of this figure:
- the glacier's basin delineation from the RGI is included in Fig.1b;
- the label "Stonebreen" was placed more to the centre of the whole basin;
- the size of the north arrows and that of the font of the scale bars were increased;
- the specifications "southern lobe" and "northern lobe" are now included Fig.1b;
- the name of the ice cap ("Edgeøyjøkulen") is now included Fig.1b.

*Fig. 2*
*The background is not contrasted enough, which makes the map difficult to read .*

The following changes were included in the new version of this figure:
- the background image is now a panchromatic one;
- the color scheme of the glacier outlines is now following a blue-to-red graded colour scheme with time;
- a scale bar was added;
- an inset map was added to show the frontal retreat and advance of the southern lobe of Stonebreen with more details.

*Fig. 5-8*
*These figures could be combined in one figure.*

We think that how these figures are combined depends on the way the paper is read. If a printed version is considered, then we agree that having all the images in one page would be an advantage. But if a digital version of the pdf is considered on a computer screen, then we think that having only two large images side by side on the same position on every page that can be alternatively viewed with PgUp and PgDn is of great advantage for understanding how velocity changes are happening in time and space and how this is related to height changes (Fig. 3).

*Fig. 10*
*Although obvious, blue and red dots should be explained in caption. Corresponding terminus position would be a must.*
Done, the colours of the dots are now explained in the caption. For the terminus positions see reply above.

---

## Author Comment (AC2) · 9 Jan 2017

**Response to Anonymous Referee #2**

Our response to the Referee #2 comments are given below and appropriate changes to the paper have been included in the new version of the manuscript. A "Track Changes" version and a "clean" version of the revised manuscript were prepared.

*General Description:*
*Strozzi et al., [2016] use optical and SAR datasets to observe the dynamic evolution of the Stonebreen Glacier of Edgeøyjøkulen Ice Cap (Svalbard) in 1994, and more frequently from 2011-2016. They combine the observed pattern of velocity fluctuation with glacier geometry evolution (determined from DEM differencing) and terminus position change, in order to speculate at the mechanisms causing the observed fluctuations in surface motion. A secondary stated goal is to evaluate the potential of frequent standard coverage acquisitions from recent earth observation missions (such as Landsat-8 and Sentinel-1) in order to analyze temporal variability in ice motion. The authors draw upon well used methods, and although there is little novel with regard to the methodology, they are nevertheless suitable for this type of work and reasonable uncertainty levels are provided. For the most part, the paper is well written and easy to follow, although in some places small typos need to be corrected and some modified work choice would help improve clarity. The tables are generally well done and complete, although some of the figures may be combined to improve clarity. At present, the discussion section of the paper is a bit too brief. I suggest that the section begin with a short paragraph which outlines how the velocity variability observed here differs significantly from glacier surging (which is observed in other basins of Edgeøyjøkulen Ice Cap), once the distinction from surging is made clear, then the other mechanisms that may be causing the variability in ice motion and the reasons for and against each of these mechanisms from the observations, can be described. I suggest that the authors also look at the "pulse" mechanism described by Van Wychen et al., [2016] for the Canadian Arctic as another mechanism that may be inducing fluctuations in ice motion. Finally, given that a major goal of this work is to assess the importance of frequent standard coverages of earth observation data for glacier velocity monitoring, there needs to a portion of the discussion devoted to this topic and more than a single sentence regarding this topic in the concluding remarks. Despite these comments, the authors now provide a much more comprehensive record of ice velocities for Stonebreen than was previously available and the dynamic behaviour observed here may apply to other glaciers in Svalbard (and other Arctic regions). I have provided a number of points below for the authors to address.*

Thank you for pointing out that we are now providing a much more comprehensive record of ice velocities for Stonebreen than was previously available and that the dynamic behaviour observed here may apply to other glaciers in Svalbard. The fact that with Landsat-8 and Sentinel-1 it is now possible to observe ice surface velocities of many Arctic glaciers at high temporal sampling is one of the major considerations we wanted to raise up with our paper. As suggested, we will make more clear in the revised version of the paper the importance of frequent standard coverages of earth observation data for glacier velocity monitoring. We will also revise the discussion following the suggestions given above - and repeated below where we will summarize the changes we have been making - to put the frontal destabilization of Stonebreen more in the general context of surges over Svalbard.

*Specific Comments*
*Minor Changes*
*PAGE 1*
*L6: Please provide a reference for the warming trend observed since the 1990s.*
This is part of the Abstract, references are usually not included here. In addition, because the 20th-century warming trend in air temperature in the Arctic has been published in the IPCC and largely reported in the news, we think that it is not necessary to provide a reference for this as it somehow

represent common knowledge.

*L7: "ice mass loss" -> "mass loss"*
Done.

*L11: "glacier's" -> "glacier"*
Done.

*L12: suggest changing "speed increases" to "velocity increases"*
Done.

*L13: Please provide references.*
Again, this is the Abstract, where references are usually not included. For references see page 2, line 16.

*L14-15: "from 1971 until 2011 followed since 2012 by a strong increase in ice surface velocity along with a decrease of volume and an advance in frontal extension" -> "from 1971 until 2011, followed by (since 2012), a strong increase in ice surface velocity along with a decrease of volume and frontal extension".*
Done.

*PAGE 2*
*L3: "The total calving flux of Svalbard is dominated by a few large and fast-flowing glaciers" please provide a reference.*
Done, Dowdeswell et al. (2008).

*L4: "So, far" -> "So far,"*
Done.

*L5: "A few glaciers" -> "A few glaciers," add comma*
Done.

*L9: "overdeepenings in the glacier bad" -> "overdeepenings of the glacier bed"*
Done.

*L10: "reduced buttressing" and "changes in the back-stressing sea ice cover in front of the glaciers" are these differing mechanisms? If so, please clarify the distinction.*
Yes, these are two differing mechanisms. One one hand, warm ocean water is reaching the fronts of the glaciers and is causing their retreat. On the other hand, sea ice cover in front of the glaciers is changing. In both cases, we have as a consequence changes in the back-stressing, i.e. reduced butressing.

*L11-12: Please provide a reference or example to back up the statement.*
References are provided just above, see Dunse et al. (2012) and Schellenberger et al. (2015), we don't want to repeat this just after one line.

*L14: "of Svalbard" -> "of the ice masses of the Svalbard Archipelago"*
Done.

*L22: "seem possible" -> "seems possible"*
Done.

*L24: "data a" -> "data, a" (add comma)*
Done.

*PAGE 3:*
*L4: Please provide a lat/long coordinate for Stonebreen.*
Done.

*L6-7: "new missions" -> "new earth observation missions" such as Sentinel-1 (SAR) and Landsat 8 (optical) to detect. . ."*
Done.

*L10: "5,073 km2" -> "5,073 km2" change to superscript.*
Done.

*L11: "The eastern side of Edgeøya is covered by the Edgeøyjøkulen Ice Cap, which had an area of 1365 km2 in 1985". . . (also note the superscript) –> please modify text.*
Done.

*L12-13: "is among the least well" -> "are among the least well", also please add the (Dowdeswell and Bamber, 1995) reference to this statement.*
Done.

*L19: "extension" -> "area"*
Done.

*L20: Mark the northern lobe of Stonebreen with an "*" and the southern lobe of Stonebreen with an "#" on Figure 1 to make it clear to the reader exactly where you are referring to. Also suggest making all the glaciers previously identified as surge-type (e.g. from Liestøl, 1993 and Dowdeswell and Bamber, 1995) with an "*".*
We now placed on Fig. 1 the label "Stonebreen" more to the center of the whole basin, which makes much clearer the distinction between the northern and southern basins without the need of any mark. Our work specifically concentrate on Stonebreen, therefore we think that marking all other previously identified as surge-type glaciers in Fig. 1 will only diverge the reader to not necessary information for the understanding of our paper. If a reader is interested to this kind of information, the appropriate references are provided.

*L7-11: Please provide an indication of the relatively uncertainty between glacier delineations versus pixel size between sensors. Have all the images been georeferenced to a common image?*
Landsat data are available orthorectified from the U.S. Geological Survey and were not co-registered to a common geometry but relative co-registration between the scenes used was checked as well as absolute georeferenced against mapped rock outcrops and non-glaciated coastlines. Only Landsat scenes that passed these visual checks were used further. All images are from the summer months with good contrast on clean ice. The relatively uncertainty of glacier delineations is on the order of one pixel, i.e. up to 30 m.

*L15: Please provide an uncertainty value for the NPI DEM.*
From a study on the nearby Digerfonna (just west of Edgeøyajokulen) we estimate the accuracy of the elevations of the NPI DEM to be on the order of few meters to around ±12m for difficult terrain (Kääb 2008).

*L23: "Digerfonna Kääb (2008)" -> "Digerfonna Ice Cap, Kääb (2008)"*
Done.

*L1-9: Please provide a description of the window sizes used for the SAR offset matching algorithm.*
Matching window sizes of 64x196, 30x120, 128x128 and 512x128 pixels were applied to the ALOS PALSAR, Radarsat-2 Wide, Radarsat-2 WUF and Sentinel-1 IWS data, respectively.

*L12: "Landscape 8" -> "Landsat 8"*
Done.

*L13: "For good visual contrast such as given for our study site and data due to the crevassed and snow-free glacier" -> "For areas of good visual contrast, such as those in our study site due to crevassed and snow-free glacier surfaces, displacement accuracies. . ." also please provide a reference for these values.*
Done.

*L1-16: For both the SAR offset tracking and the optical feature tracking, please describe how mismatches or blunders are removed from the dataset. Was this completed manually? Was the strength of the cross-correlation value used to flag poor matches? Please describe in more detail.*
For both SAR and optical tracking mis-matches or blunders were filtered by applying a threshold on the correlation coefficient, by iteratively discarding matches based on the angle and size of displacement vectors in the surrounding area, and by using a high-pass filter on the resulting displacement fields.

*L20: "on a Landsat image of 14/07/2014" this can be removed as it appears in the figure caption.*
Done.

*PAGE 7:*
*L2: "NPI DEM of 1990" -> "NPI DEM of 1970"*
Done.

*L3: "From 1990" -> Please check here and throughout the manuscript and figures. You note that the NPI DEM is derived from aerial photography in 1970 (Page 4, L16), however at other times in the manuscript (see section 4.2 you describe it as being from 1990). Please correct.*
Done.

*L2-4: This section is somewhat awkwardly phased and could be clarified and would benefit from further description. Please describe more fully what "height losses of up to 150 m over current sea level and up to 100 m over current ice" really means. Suggest changing "current" to the last year when DEM data is available. It is noted that height changes of 100-150 m are observed, however the Figure scaling only shows elevation changes +/- 50 m, please adjust the scale bar so that the description in texts describes what can fully be seen in the figure.*
This sentence was reformulated. "From 1990 to 2010/2012 we observe a general pattern of height losses along the coast of Stonebreen. Over the southern lobe the ice surface losses were up to 150 m along the coast."
The Fig. 3a scale bar was selected to match that of Fig. 3b, to best infer the large pattern of ice surface losses along the coast, and to highlight the slightly increasing elevation changes at higher sections (see discussion in Section 5.5). We agree that adjusting the scale bar to larger values will permit better distinction between different rates of ice surface losses, but our choice is not to change the scale bar of this figure for the reasons mentioned above.

*L13: "The velocity is lower towards south" -> "The velocity is lower towards the south"*
Done.

*L14: "The northern sector is decorrelated, i.e. flowing faster" -> Yes, this can cause decorrelation, but what about change in the glacier surface that could cause decorrelation? Provide further evidence why you attribute it to faster flow speeds rather than changes in surface characteristics.*

Please see the following sentence, where ALOS PALSAR results of the same year are discussed: "ALOS PALSAR data of the same period as ERS-2 SAR (Fig. 5a) quantifies the ice surface velocity in the northern to maximum 300 m/yr."

*L17: "are indicating" -> "indicate"*
Done.

*L19: "in summer of 2014 (a) respectively 2015 (b)" -> awkwardly phrased, should be re-written.*
Done.

*L20: "with a migration of the front of increased speeds towards inland" -> "due to an inland migration of a front of elevated velocities"*
Done.

*L20: "1'500" -> "1,500" or just "1 500"*
Done.

*L24: "12-days" -> "12-day"*
Done.

*PAGE 8:*
*L5-6: "dynamically active sector is increasing again inland" -> does this mean that the dynamically active sector is again migrating inland?*
Yes, this sentence was reformulated.

*L13: "2'500" -> "2,500" or "2 500"*
Done.

*L14: "The different SAR and optical satellite sensors complement each other very well".*
*In principal I agree with this statement, however I would like to see the authors develop this idea further, especially because evaluating the potential of frequent standard acquisitions is stated as a secondary goal of this paper (Page 3, lines 6-8). To further illustrate this statement, I suggest creating a timeline figure that shows all the image acquisition broken down (colour coded) by sensor for the period from ~2010-2016. For an example, see bottom panel of Figure 2 in Burgess et al., [2012] of how this can be accomplished. I recognize that this information is available in Table 2, however the visual timeline would show the ready more easily how much of the time during the study period that the site was under observation, and further highlight the point that frequent observations improve our understanding of the temporal evolution of glacier velocities. This newly created figure may have the potential to replace Table 2, or at least move that table to supplementary materials.*

The potential of frequent standard acquisitions is now further developed in the conclusions of the paper (see below for further details).

Figure 9 is actually already a timeline figure that shows all the image acquisitions colour coded by sensor. In this figure we can indeed graphically easily see "how much of the time during the study period the site was under observation, and further highlight the point that frequent observations improve our understanding of the temporal evolution of glacier velocities."

We agree that Table 2 could be added as supplemental material.

*L20: "increase in slope" -> "increase in surface slope"*
Done.

*L22: "increment" -> "increase"*
Done.

*PAGE 9:*
*L7-8: "The increased elevation loss towards the front lead to an increase" -> "The elevation loss at the glacier front between the NPI DEM and the IDEM led to an increase in surface slope of ∼2°..."*
Done.

*L18-20: This sentence is somewhat awkward to me and should be rewritten to improve clarity.*
This sentence was reformulated.

*PAGE 10*
*L10-12: The method for extracting and comparing backscatter intensity needs to be more fully described, and this should be provided in the methods section. Have all the backscatter intensity values been corrected to sigma nought values to account for various incident angles to enable comparison from different acquisitions and incidence angles? Or can this be neglected because all of the images are interferometric pairs with the same viewing geometry? This is not clearly described in text and should be. In addition to only using the backscatter values to determine melt rates, is it possible to use nearby meteorological station data or NCEP reanalysis to strengthen your claims?*
Agree, a few further details and a reference were added here to give more information about the method. We prefer however not to add a new chapter in Section 3 because this analysis is only marginal for our work.
We also agree that meteorological station data or NCEP reanalysis would help support our claims as well. However, we are not aware of nearby meteorological station data and NCEP reanalysis was not considered in our analysis. Here, we wanted to highlight the potential of the SAR backscattering intensity to spatially map in a quantitative way the surface melt-water.

*PAGE 11*
*L13-17: This portion of the paragraph is somewhat unclear and could be tidied to improve clarity.*
Done.

*PAGE 12:*
*L1-5: These sentences can be modified to improve clarity.*
Done.

*L10: suggest changing "(surge-type?) instability" to just "instability"*
Done.

*L15-L19: Comparisons with unpublished data for Basin-2 should be presented at the end of the discussion section rather than being introduced within the conclusions.*
We prefer to keep this comparison in the conclusions as a kind of outlook, because the discussion there is extending our findings over Stonebreen on a broader scale and is relating our specific work on a single glacier to the more general context of glacier destabilization over Svalbard.

*L21: "at high temporal sampling", suggest quantifying this remark. How often will Svalbard be*

*covered with standard acquisitions in the future? Every 12 days going forward? Or does it change seasonally? Provide a bit more information here.*
Done, we specified even up to every 6 days with Sentinel-1A and B.

*PAGE 13:*
*L4: "The Reearch Council of Norway" -> "The Research Council of Norway"*
Done.

*PAGES 13-16:*
*Check all references, in some cases DOI numbers are missing.*
Done.

*Substantive Comments:*
*Methods/Results Sections*
*Please further discuss the comparison of Sentinel-1 backscatter values over the melt season. Currently, this topic only appears in the discussion section, but should also be described in the methods and results sections.*
A few further details and a reference were added in the Discussion to give more information about this method. We prefer however not to add a new chapter in Section 3 because this analysis is only marginal for our work and does therefore not belong to the main methods.

*Discussion Section*
*I would like to see the discussion section begin with a brief description of why the observed velocity pattern does/does not conform to traditional surge theory, which then narrows down to introduce the alternate processes provided by the authors that could explain the velocity variability. One potential mechanism that is not described by the authors, but may be relevant, is "pulsing" which has been observed in other Arctic regions (see Van Wychen et al., 2016). This mechanism involves geometry changes, glacier advance and glacier speed-up, and the authors may want to include this as another potential mechanism in their discussion.*

In order to put our work on the frontal destabilization of Stonebreen more in the general context of glacier surges we first included a few more information about the mechanism of glacier surges over Svalbard in the Introduction.
"Surge-type glaciers undergo a cyclic behaviour, with periods of rapid acceleration and advance (active phase) followed by periods of slow flow where ice fluxes are less than balance fluxes (quiescent phase) (Clarke, 1987). In a typical Svalbard glacier surge cycle the surge starts with a years-long period of steady acceleration, followed by a months-long period of relatively rapid acceleration, a length of the active phase of typically 3-10 years, and a very gradual end of the fast flow phase with velocity decreasing over a years-long period (Murray et al., 2003). The long active (~7-15 years) and quiescent (~50-100 years) phases, combined with surge termination that occurs over a multi-year period and velocity changes between the two phase of one or two orders of magnitude, suggest that the Svalbard type surges are linked to changes in basal thermal conditions rather than subglacial water pressure (Murray et al., 2003)."

Then, we briefly discussed at the beginning of Section 5 why the observed velocity pattern over Stonebreen does not conform to traditional surge theory for glaciers over Svalbard.
"Over the southern lobe of Stonebreen we observe a slow steady retreat of the glacier from 1971 to 2011 followed since 2012 by a strong increase in ice surface velocity with prominent seasonal variations along with a decrease of volume and an advance in frontal extension. The acceleration phase of the southern lobe of Stonebreen was lasting at least 3 years, more than the months-long period of relatively rapid velocity increase of a typical Svalbard glacier surge cycle (Murray et al., 2003), and the acceleration was not constant but seasonally modulated. So far, no deceleration

phase is observed over the glacier."

Finally, we expanded the conlusions to include also comparison to glacier surges or pulses in other regions as Patagonia (see Mouginot and Rignot, 2015) and the Canadian Arctic (see Van Wychen et al., 2016).
"Also other glaciers in different regions, such as Pío XI in Patagonia (Mouginot and Rignot, 2015), presents similar features as Stonebreen, such as shallow bed below sea level at the terminus, large thickness changes, strong seasonal and annual variations, and large melt water production. Over the Canadian Arctic Van Wychen et al. (2016) introduced the concept of pulse-type glaciers. Over this kind of glaciers the velocity variability initiates in and propagates upglacier from the lowermost sections of the glacier near the terminus and is largely restricted to regions where the bed lies below sea level. Even if also for Stonebreen the instability initiated near the terminus, the velocity variability is now migrating upglacier more than the 6km inland from the 2014 front where the glaciers is believed to be grounded."

*Given that the stated secondary goal of the paper is to evaluate the potential of frequent standard coverages of earth observation data to analyse glacier dynamics there needs to be a portion of the discussion section devoted to this topic. Currently, the discussion section does not provide any reason why the authors believe that frequent standard coverages are beneficial beyond a very brief statement. Although this may seem somewhat obvious, it needs to be discussed fully if the authors intend on it being a major outcome of the paper.*
This is now discussed in the conclusions.

*Conclusion Section*
*Again, given that the secondary goal of the paper is to show how beneficial it is to have frequent coverage of earth observation datasets for glacier monitoring, the conclusions should devote more than one sentence to this topic. Specifically, the authors should note that with the data they had available, that they were able to monitor the dynamic evolution of this glacier nearly continuously from ∼2011-2016, and that this likely would not have been possible in the recent past. The authors may also want to speculate as to how recently launched (Sentinel 1b) or future sensor (Radarsat Constellation Mission) will even further increase the amount of data available for this type of monitoring.*
Done. The fact that with Landsat-8 and Sentinel-1 it is now possible to observe ice surface velocities of many Arctic glaciers at high temporal sampling is one of the major considerations we wanted to raise up with our paper. As suggested, we made more clear in the revised version of the conclusion the importance of frequent standard coverages of earth observation data for glacier velocity monitoring.

*FIGURES:*
*Figure 1: needs to be modified and clearly indicate that the ice cap is named "Edgeøyjøkulen" and that "Stonebreen" is a glacier basin within the Edgeøyjøkulen Ice Cap. Suggest adding the glacier basin delineations from the GLIMS Randolph Glacier Inventory and provide an arrow to the Stonebreen Glacier Basin. Suggest also adding a notation, such as "*" to the basins that have previously been identified as "surge type" in the literature. Increase the size of the north arrows as well as the font of the scale bars (particularly on (b), (c), (d)) to improve readability.*
The following changes were included in the revised version of this figure:
- the glacier's basin delineation from the RGI is included in Fig.1b;
- the label "Stonebreen" was placed more to the centre of the whole basin;
- the size of the north arrows and that of the font of the scale bars were increased;
- the specifications "southern lobe" and "northern lobe" are now included Fig.1b;
- the name of the ice cap ("Edgeøyjøkulen") is now included Fig.1b.
On the other hand, because our work specifically concentrate on Stonebreen, we think that marking

all other previously identified surge-type glaciers in Fig. 1 will only diverge the reader to not necessary information for the understanding of our paper. If a reader is interested to this kind of information, the appropriate references are provided.

*Figure 2: Please provide the background image as a panchromatic image rather than a multi-spectral image, right now the image appears washed out and for clarity would appear better as a grayscale background image. Suggest changing the colour scheme of the glacier outlines and use a graded colour scheme (blue to red with time) rather than a mixture of colour and gray outlines. Please add a scale bar to this figure as it will aid the reader to determine the scale of terminus position change along the calving front. Suggest adding an inset map to show the frontal advance of the southern lobe of Stonebreen between 2011 and 2015, at the current scale it is difficult to see.*
The following changes were included in the new version of this figure:
- the background image is now a panchromatic one;
- the colour scheme of the glacier outlines is now following a blue-to-red graded colour scheme with time;
- a scale bar was added;
- an inset map was added to show the frontal retreat and advance of the southern lobe of Stonebreen with more details.

*Figure 4: It would be more beneficial if these figures were projected to velocities rather than presented as interferomgrams. This would enable comparison of glacier velocities shown in Figures 5-8 and may be beneficial to readers that are not familiar with interpreting interferometric fringes.*
See Section 3.3: "Phase unwrapping to obtain displacement values (Werner et al., 2002) was not attempted because undersampling of the SAR data in relationship to the rate of movement is easily causing phase unwrapping errors, similar to what is observed in the case of mining". For readers that are not familiar with interpreting interferometric fringes we are providing at the beginning of Section 4.3 very extensive explanations.

*Figures 4-8: The authors should consider combining Figures 4-8 into a single figure with multiple panes and with a common glacier velocities scaled colour bar. By combining these figures together it would help the reader understand the dynamic evolution of the glacier more clearly. Also note, in figures 5-8, that the glacier velocity colour bar and the velocities provided on the map are fully saturated at the high end of the velocity bar, please consider increasing the colour bar scale to provide more distinction between velocity bands.*
We think that how these figures are combined depends on the way the paper is read. If a printed version is considered, then we agree that having all the images in one page would be an advantage. But if a digital version of the pdf is considered on a computer screen, then we think that having only two large images side by side on the same position on every page that can be alternatively viewed with PgUp and PgDn is of great advantage for understanding how velocity changes are happening in time and space and how this is related to height changes (Fig. 3).
Colour saturation in these images was carefully selected in order to illustrate in the best way the spatial distribution of the instability. If the colour bar is increased then less distinction between slow and fast moving areas is provided. The colour bar of Fig. 5 is different in order to better highlight where the frontal instabilities started with lower velocities.

*Figure 10: It may be beneficial to add a trend line for both data series which shows that RADAR backscatter values decrease as glacier velocities increase (albeit with some temporal lag) to indicate that melt may be modulating ice flow. Also, the figure caption needs to be more description, e.g. it needs to say that the blue marking indicate backscatter values and that red markings indicate ice surface velocities.*
We also prepared an image with trend lines, both we found out that it was less clear than the one without trend lines. What the blue and red dots are is now explained in the caption.

*REFERENCES:*

*Burgess, E.W., Forster, R.R., Larsen, C.F., Braun, M. [2012], Surge Dynamics on Bering Glacier, Alaska, in 2008-2011. The Cryosphere, 6, 1251-1262, doi: 10.5194/tc-6-1251-2012.*
This reference is not included, because Figure 9 is already somehow a timeline figure that shows all the image acquisitions colour coded by sensor.

*Van Wychen, W., Davis, J., Burgess, D.O., Copland, L., Gray, L., Sharp, M., and Mortimer, C. [2016], Characterizing interannual variability of glacier dynamics and dynamic discharge (1999-2015) for the ice masses of Ellesmere and Axel Heiberg Islands, Nunavut, Canada. Journal of Geophysical Research: Earth Surface, 121, doi:10.1002/2013JF003839.*
This reference is now included and discussed.

---

## Author Response (AR2)

**Response to Editor Review**

*Thanks for this new version of your manuscript and the point to point reply to both reviews. I am happy to accept your paper for publication in The Cryosphere. I have nevertheless few minor comments that should be accounted for before final publication (see below).*

Thank you for accepting our paper for publication in The Cryosphere and for providing the additional minor comments. We addressed these valuable comments in the latest revion of the manuscript. Our response to the Editor Review is given below, a "Track Changes" and a "clean" version of the revised manuscript were uploaded to our File Manager.

*Comments:*
*page and line refer to the paper version with highlighted changed.*

*- page 1, line 20: the contribution of glaciers and ice-sheets is closer to 2/3 of the current total SLR than 1/3?*

According to Table 13.1 of Church et al. (2013) the contribution of glaciers and ice-sheets to the current total SLR is closer to 1/3 than 2/3.

*- page 2, line 13 and at other places in the manuscript: pin-point should be pinning-point?*

Done.

*- page 2, line 24: the total glacier mass balance of of the ice masses of the Svalbard Archipelago -> the total glacier mass balance of the Svalbard Archipelago*

Done.

*- page 3, line 17: earth -> Earth*

Done.

*- page 5, line 6 (and elsewhere in the manuscript, check this): 1:100'000 -> 1:100 000*

Done.

*- Table 2: I still think that this long table should be moved in the supplementary material and synthetic informations (number of maps and time interval) given after the introduction of each instrument.*

Done.

*- page 6, line 1 (and elsewhere in the manuscript, check this): 3-days -> 3-day*

Done.

*- page 6, line 3: don't -> do not*

Done.

*- page 6, line 5, pair is only 15 m the phase -> pair is only 15 m, the phase*

Done.

*- Legend of Fig. 3 and elsewhere in the text: not sure that coastal line is correct. Isn't it coast line ?*

Coastal line was changed to coastal outline.

*- page 10, line 3: the sentence "So far, no deceleration phase is observed over the glacier" is a bit in contradiction with what is said elsewhere as you have observed lower winter velocity since 2015? May be you should elaborate a bit more about what you want to said here?*

Agreed, we changed this to "So far, no distinct deceleration phase is observed over the glacier."

*- page 10, line 21: are lower than the surface speeds -> are lower than the increase in surface speeds*

Done.

*- page 10, line 24: in ice deformation from -> in deformational velocity from (because the value you give after are velocity and not deformation).*

Done.

*- page 13, lines 10 - 16: 1'200-> 1 200 ; km^3: delete '^' and 3 should be an exponent of km*

Done.

*- page 13, line 20: the total mass loss of Edgeøyjøkulen should be given in the same units as the value for Stonebreen (km3/yr instead of Gt/yr) to allow an easy comparison.*

Done, Gt/yr is no more used.

**Supplementary Table S1. Sensors, acquisition dates and time intervals of the satellite image pairs considered for ice surface velocity estimation.**

| Satellite Sensor | Acquisition Date 1 | Acquisition Date 2 | Time Interval |
|---|---|---|---|
| ERS-1 SAR | 02/01/1994 | 05/01/1994 | 3 days |
| ERS-2 SAR | 22/03/2011 | 25/03/2011 | 3 days |
| ALOS PALSAR Fine Beam Single | 14/11/2010 | 14/02/2011 | 92 days |
| | 04/01/2011 | 19/02/2011 | 46 days |
| Radarsat-2 Wide | 09/02/2009 | 05/03/2009 | 24 days |
| | 05/03/2009 | 29/03/2009 | 24 days |
| | 29/03/2009 | 22/04/2009 | 24 days |
| | 04/06/2010 | 28/06/2010 | 24 days |
| | 06/05/2011 | 23/06/2011 | 48 days |
| | 23/06/2011 | 17/07/2011 | 24 days |
| | 17/07/2011 | 10/08/2011 | 24 days |
| | 10/08/2011 | 03/09/2011 | 24 days |
| | 03/09/2011 | 27/09/2011 | 24 days |
| | 27/09/2011 | 21/10/2011 | 24 days |
| | 21/10/2011 | 08/12/2011 | 48 days |
| | 13/03/2012 | 06/04/2012 | 24 days |
| | 06/04/2012 | 30/04/2012 | 24 days |
| | 30/04/2012 | 24/05/2012 | 24 days |
| | 24/05/2012 | 17/06/2012 | 24 days |
| | 17/06/2012 | 04/08/2012 | 48 days |
| | 04/08/2012 | 28/08/2012 | 24 days |
| | 28/08/2012 | 21/09/2012 | 24 days |
| | 21/09/2012 | 15/10/2012 | 24 days |
| | 15/10/2012 | 08/11/2012 | 24 days |
| | 08/11/2012 | 02/12/2012 | 24 days |
| | 12/02/2013 | 08/03/2013 | 24 days |
| | 08/03/2013 | 25/04/2013 | 48 days |
| | 25/04/2013 | 19/05/2013 | 24 days |
| | 12/06/2013 | 30/07/2013 | 24 days |
| | 30/07/2013 | 23/08/2013 | 24 days |
| | 23/08/2013 | 16/09/2013 | 24 days |
| | 16/09/2013 | 10/10/2013 | 24 days |
| | 10/10/2013 | 03/11/2013 | 24 days |
| | 27/11/2013 | 21/12/2013 | 24 days |
| | 21/12/2013 | 07/02/2014 | 48 days |
| Landsat 8 | 14/07/2014 | 06/08/2014 | 23 days |
| | 06/08/2014 | 24/08/2014 | 18 days |
| | 24/08/2014 | 31/08/2014 | 7 days |
| | 06/07/2015 | 02/08/2015 | 27 days |
| | 02/08/2015 | 18/08/2015 | 16 days |
| | 18/08/2015 | 17/09/2015 | 30 days |
| | 26/06/2016 | 28/07/2016 | 32 days |
| Radarsat-2 Wide Ultra Fine | 04/02/2016 | 28/02/2016 | 24 days |
| | 28/02/2016 | 23/03/2016 | 24 days |
| Sentinel-1 Interferometric Wide Swath | 21/01/2015 | 02/02/2015 | 12 days |
| | 02/02/2015 | 14/02/2015 | 12 days |
| | 13/08/2015 | 25/08/2015 | 12 days |
| | 25/08/2015 | 06/09/2015 | 12 days |
| | 06/09/2015 | 18/09/2015 | 12 days |
| | 18/09/2015 | 30/09/2015 | 12 days |
| | 30/09/2015 | 12/10/2015 | 12 days |
| | 12/10/2015 | 24/10/2015 | 12 days |
| | 24/10/2015 | 05/11/2015 | 12 days |
| | 05/11/2015 | 17/11/2015 | 12 days |
| | 17/11/2015 | 29/11/2015 | 12 days |
| | 23/12/2015 | 04/01/2016 | 12 days |
| | 28/01/2016 | 09/02/2016 | 12 days |
| | 09/02/2016 | 21/02/2016 | 12 days |
| | 21/02/2016 | 04/03/2016 | 12 days |
| | 04/03/2016 | 16/03/2016 | 12 days |

| | | |
|---|---|---|
| 16/03/2016 | 28/03/2016 | 12 days |
| 28/03/2016 | 09/04/2016 | 12 days |
| 09/04/2016 | 21/04/2016 | 12 days |
| 21/04/2016 | 03/05/2016 | 12 days |
| 03/05/2016 | 15/05/2016 | 12 days |
| 15/05/2016 | 27/05/2016 | 12 days |
| 27/05/2016 | 08/06/2016 | 12 days |
| 08/06/2016 | 02/07/2016 | 24 days |
| 02/07/2016 | 14/07/2016 | 12 days |
| 14/07/2016 | 26/07/2016 | 12 days |
| 26/07/2016 | 07/08/2016 | 12 days |
| 07/08/2016 | 19/08/2016 | 12 days |
| 19/08/2016 | 31/08/2016 | 12 days |